# FLSL: Feature-level Self-supervised Learning

**Qing Su**[1][*] **Anton Netchaev**[2]**, Hai Li**[3]**, and Shihao Ji**[1]
[1]Georgia State University, [2]U.S. Army ERDC, [3]Duke University

## Abstract

Current self-supervised learning (SSL) methods (*e.g.*, SimCLR, DINO, VICReg, MOCOv3) target primarily on representations at instance level and do not generalize well to dense prediction tasks, such as object detection and segmentation. Towards aligning SSL with dense predictions, this paper demonstrates for the first time the underlying *mean-shift* clustering process of Vision Transformers (ViT), which aligns well with natural image semantics (*e.g.*, a world of objects and stuffs). By employing transformer for joint embedding and clustering, we propose a bi-level feature clustering SSL method, coined Feature-Level Self-supervised Learning (FLSL). We present the formal definition of the FLSL problem and construct the objectives from the *mean-shift* and *k*-means perspectives. We show that FLSL promotes remarkable semantic cluster representations and learns an encoding scheme amenable to *intra-view* and *inter-view* feature clustering. Experiments show that FLSL yields significant improvements in dense prediction tasks, achieving 44.9 (+2.8)% AP and 46.5% AP in object detection, as well as 40.8 (+2.3)% AP and 42.1% AP in instance segmentation on MS-COCO, using Mask R-CNN with ViT-S/16 and ViT-S/8 as backbone, respectively. FLSL consistently outperforms existing SSL methods across additional benchmarks, including UAV object detection on UAVDT, and video instance segmentation on DAVIS 2017. We conclude by presenting visualization and various ablation studies to better understand the success of FLSL. The source code is available at `https://github.com/ISL-CV/FLSL`.

## 1 Introduction

Following its success in natural language processing (NLP) [47, 5, 20], self-supervised learning (SSL) with transformer [58, 22] has emerged as a highly effective strategy and a popular model choice over the CNN-based counterparts in vision tasks. The remarkable performance achieved by SSL has been demonstrated by SimCLR [14], MOCOv3 [16], DINO [10], VICReg [3], SwAV [9], BYOL [27], and among others. Without relying on manual supervision, a successful paradigm of SSL promotes semantic representations conducive to the downstream tasks, *e.g.*, classification, detection and segmentation. However, most existing SSL methods operate at the instance-level, where an encoder is trained to maximize the agreement of the representations of multiple augmented views of an image. Though demonstrating strong performance on the classification tasks [14, 29], the instance-level SSL is inherently misaligned with the dense prediction tasks, such as object detection, where the lower level semantic information plays a bigger role than the instance-level semantic information. This leads to inferior transferability to those dense prediction tasks.

Recent attempts to bridge the semantic gap are mainly based on region [50], patch [69, 21], or pixel (*i.e.*, dense feature) matching tasks [63, 73, 38] with optional instance-level objectives. However, learning of distinct representation for each image patch or region still mismatches the natural semantics within an image (referred to as local semantics), where features of the same semantics should be highly correlated other than being distinct. Semantics can range from features of high similarity, features of the same object, to more complex semantic structures. In light of this, methods such as SoCo [65], ORL [70] and DetCon [32] leverage the off-the-shelf algorithms, *e.g.*, *selective*

---

[*]To whom correspondence should be addressed: `qsu3@gsu.edu`

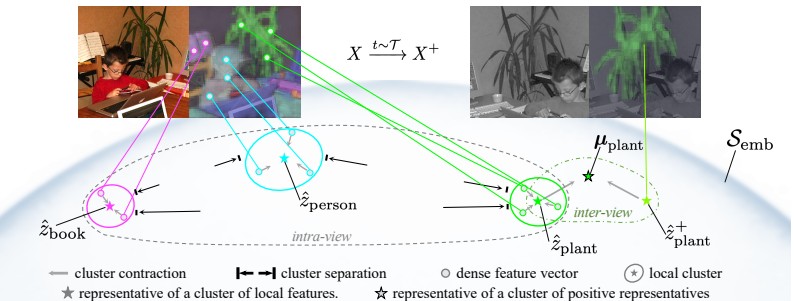

Figure 1: The bi-level clustering of FLSL. An object or stuff in an image is essentially a cluster of features. Hence, their representations can be extracted as cluster representatives, *e.g.*, modes. In FLSL, we aim to make these representations both locally and globally semantic via a bi-level clustering process. On the first level, the ***locally*** semantic representations are fostered by driving features of various concepts (book, person, plant, etc.) closer to their cluster modes $\hat{z}_{cls}$s and far away from features of other concepts within an image(*intra-view clustering*). On the second level, cluster modes serving as representations $\hat{z}_{cls}$s are pushed closer to their positive samples $\hat{z}_{cls}^+$s in $X^+$, which is augmented via a random transformation $t \sim \mathcal{T}$ (*inter-view clustering*). In such a way, those representations encode the same category information and become ***globally*** semantic.

*search* [55] and *Felzenszwalb-Huttenlocher* algorithm [25] to impose the semantic constraint to the contrastive learning pipeline. Nonetheless, the inclusion of a non-trainable region proposal module in those methods restricts the model's ability to learn the distinct representations for those RoIs from the rest of an image. This ability is vital in representation learning for object detection.

Existing SSL methods targeting dense prediction primarily focus on the learning of globally semantic representations of image sub-regions, such as RoIs, patches, or pixels. However, these methods fall short with limited consideration for the alignment of those representations with local semantics. This observation leads us to ask the following question: Can we learn a representation that is both locally and globally semantic for a group of features (*e.g.*, representing an object or stuff) in an end-to-end trainable SSL approach? To this end, we propose the *Feature Level Self-supervised Learning* (FLSL). It leverages the *mean-shift* clustering process inherent in transformer to extract modes as representations and incorporates *k-means*-based SSL approach to ensure that the extracted representations are semantically coherent both locally and globally. Figure. 1 illustrates an overview of FLSL with details to be discussed in Sec. 4.

**Contributions** This paper takes a step forward to bridge the gap between the current SSL methods and downstream dense prediction tasks. Our contributions are summarized as follows:

1. We demonstrate for the first time the connection between the attention mechanism and *mean-shift* clustering, and reinterpret vision transformer from the perspective of *mean-shift*.

2. By employing transformer for joint embedding and feature clustering, we propose FLSL, an end-to-end trainable SSL method that promotes the representations of feature clusters to be semantic at two levels: (i) intra-view: within an image, and (ii) inter-view: over an entire dataset.

3. The derivation and construction of the FLSL objectves is rooted in *mean-shift* and the non-empty *k-means* clustering. Semantic representations on the first level are encouraged by optimizing the intra-cluster affinity with a self-attention layer, while the second-level semantic representations are fostered via non-empty *k-means* clustering with positive samples retrieved through a cross-attention layer.

4. We validate the synergy between FLSL and ViT, and show significant improvement in transferability of the learned features to dense prediction tasks, including object detection and semantic segmentation. FLSL-pretrained ViT on ImageNet-1k (IN1k) demonstrates superior performance compared to the state-of-the-art ADCLR-IN1k [76] and MAE [40] pretrained counterparts. Moreover, it consistently outperforms existing SSL methods across additional benchmarks, including UAV object detection on UAVDT, and video instance segmentation on DAVIS 2017.

## 2 Related work

**SSL for dense prediction** Recent attempts to bridge the gap between common SSL and dense prediction tasks focus primarily on sub-region matching tricks. For example, DenseCL [63] applies contrastive learning on pairs of patches with highest similarity. However, the patch-matching trick leads to distinct representations with low correlation among patches, which is not well-suited for the semantics of a natural image. On top of the instance-level objective, PixPro [73] and LC-loss [35] factor in agreement between positive pixel pairs which are assigned through thresholded-distance in

PixPro and position projection in LC-loss. ReSim [68] maximizes the agreement between sliding-window-pooled representations in the overlapped region of two augmented views. DetCo [69] further incorporates instance-patch level contrastive losses along with instance level and patch level losses. To learn representations at object level, SoCo [65] and ORL [70] employ *selective search* to crop out RoIs. ORL further enables inter-object representation learning via BYOL [27] using top-ranked RoI pair. In contrast, SCRL [50] relaxes the semantic constraint using random crops within the intersection area of augmented views as RoIs. As discussed in Sec. 1, all of these methods focus on learning globally semantic representations for image sub-regions, and do not touch on local semantics that are necessary for dense prediction.

**Self-supervised vision transformer** In pioneering works, self-supervised training of transformer for vision tasks generally follow the paradigm of masked autoencoder in NLP [47, 20]. For instance, iGPT [13] features reconstruction of masked pixels as one of its objectives. In general, SSL for ViT can be classified into two categories: the joint-embedding strategy epitomized by DINO [10] and MoCov3 [16], and the generative approaches represented by MAE [28]. The crossover of the two strategies is demonstrated by iBOT [78]. Regarding **dense prediction**, EsViT [38], designed for Swin Transformer [43], follows the region-matching strategy and applies the DINO loss to the probabilities of positive pairs determined by highest similarity. Instead of finding the best-matching patch, SelfPatch [75] considers the direct neighbors as its positive patches. However, with limited semantics contained in a fixed small area (*e.g.*, 8-connected neighbors), the method still suffers from semantic misalignment. To address the sub-region mismatch issue of DINO, ADCLR [76] constructs query tokens from random sub-regions and treats them as extra class tokens in the DINO objective. This promotes region-aware semantic representations that better aligned with the local semantics, and leads to substantial improvement in dense prediction.

# 3   Intuition: the connection between mean-shift and attention

As discussed in Sec. 1, the misalignment between the current SSL methods and dense prediction tasks lies in the clustering bias at the semantic level. Instead of setting a fixed granularity, such as instance-level or fix-sized patch-level, a desired semantic representation scheme should be able to represent from a single patch to a cluster of patches or even an entire image. The representation space of an image can be considered as an empirical probability density function of features, and the modes (local maxima) therefore can be regarded as the representatives of clusters [11, 17, 18]. These modes can then be readily retrieved via clustering algorithms, particularly, non-parametric *kernel density estimation* (KDE) methods [62] when the image composition (*e.g.*, number of objects and stuffs) is unknown. One typical KDE-based method is the *mean-shift* clustering [33]. In the following, we first give an overview of self-attention (SA) mechanism of transformer and the mean-shift algorithm. We then show that the mean-shift update rule conforms to the SA mechanism of transformer.

**Attention mechanism** First introduced to recurrent neural networks as a context extractor for machine translation [2], attention has premised major breakthroughs in NLP with the emergence of transformer that relies solely on the *scaled dot-product* attention mechanism [58] given by

$$\text{attention}\left(\boldsymbol{Q}, \boldsymbol{K}, \boldsymbol{V}\right) = \boldsymbol{V}\text{softmax}\left(\boldsymbol{Q}^\top \boldsymbol{K} / \sqrt{D_{qk}}\right), \tag{1}$$

where $\boldsymbol{Q}$, $\boldsymbol{K}$ and $\boldsymbol{V}$ denote query, key and value matrices which pack together sets of query, key and value vectors, respectively. $D_{qk}$ denotes the dimension of query and key vectors, and $\text{softmax}\left(\boldsymbol{Z}\right)_{ij} = \exp(\boldsymbol{Z}_{ij})/\sum_k \exp(\boldsymbol{Z}_{ik})$. As a special case of attention, SA matches a sequence $\boldsymbol{Z}$ with itself to extract the semantic dependencies among its components, *i.e.*, $\boldsymbol{Q} = \mathbf{W}_Q \boldsymbol{Z}, \boldsymbol{K} = \mathbf{W}_K \boldsymbol{Z}, \boldsymbol{V} = \mathbf{W}_V \boldsymbol{Z}$, where the projections $\mathbf{W}\_$'s are the parameter matrices.

**Mean-shift clustering and attention** Given $N$ data points $\{\boldsymbol{z}_i\}_{i=1}^N \subset \mathbb{R}^D$, the kernel density estimate of $p(\boldsymbol{z})$ with kernel $K(t)$ can be defined as

$$p(\boldsymbol{z}) = \sum_{i=1}^N p(\boldsymbol{z}_i)p(\boldsymbol{z}|\boldsymbol{z}_i) = \sum_{i=1}^N \pi_i \frac{1}{T_i} K(d(\boldsymbol{z}, \boldsymbol{z}_i; \boldsymbol{\Sigma}_i)), \tag{2}$$

where $p(\boldsymbol{z}_i) = \pi_i$ is the mixing proportion of point $\boldsymbol{z}_i$, $s.t. \sum_{i=1}^N \pi_i = 1$, $T_i$ denotes the normalization term dependent only on the covariance matrix $\boldsymbol{\Sigma}_i$, *e.g.*, for a Gaussian kernel $T_i = |2\pi\boldsymbol{\Sigma}_i|^{1/2}$ and $d(\boldsymbol{z}, \boldsymbol{z}_i; \boldsymbol{\Sigma}_i) = (\boldsymbol{z} - \boldsymbol{z}_i)^T \boldsymbol{\Sigma}_i^{-1} (\boldsymbol{z} - \boldsymbol{z}_i)$ is the *Mahalanobis* distance. Finding the modes of $p(\boldsymbol{z})$ is

to seek stationary points by equating the gradient of $p(\boldsymbol{z})$ to zero, $\partial p(\boldsymbol{z})/\partial \boldsymbol{z} = 0$, which arrives at

$$\hat{\boldsymbol{z}} = \boldsymbol{f}\left(\boldsymbol{z}\right) = \sum_{i=1}^{N} p(\boldsymbol{z}_i|\boldsymbol{z})\boldsymbol{z}_i, \quad \text{with } p(\boldsymbol{z}_i|\boldsymbol{z}) = \frac{\pi_i \frac{1}{T_i} K'(d(\boldsymbol{z}, \boldsymbol{z}_i; \boldsymbol{\Sigma}_i))\boldsymbol{\Sigma}_i^{-1}}{\sum_{j=1}^{N} \pi_j \frac{1}{T_j} K'(d(\boldsymbol{z}, \boldsymbol{z}_j; \boldsymbol{\Sigma}_j))\boldsymbol{\Sigma}_j^{-1}}, \quad (3)$$

where $K' = dK/dt$. The above fixed-point iterative scheme is the *mean-shift* algorithm. Practically, on $\ell_2$-normalized vectors, for a homoscedastic *Gaussian* kernel with constant mixing proportion and isotropic covariances (*e.g.*, $\pi_i = 1/N$, $1/\sigma^2 = \tau$), Eq. 3 further simplifies to

$$\hat{\boldsymbol{z}} = \text{meanshift}(\boldsymbol{z}, \tau) = \sum_{i=1}^{N} \frac{\exp\left(\tau \boldsymbol{z}^\top \boldsymbol{z}_i\right)}{\sum_{j=1}^{N} \exp\left(\tau \boldsymbol{z}^\top \boldsymbol{z}_j\right)} \boldsymbol{z}_i \implies \hat{\boldsymbol{Z}} = \boldsymbol{Z} \,\text{softmax}\left(\tau \boldsymbol{Z}^\top \boldsymbol{Z}\right), \quad (4)$$

which conforms to the attention function (Eq. 1) with identity projection matrices, *i.e.*, $\boldsymbol{W}_Q = \boldsymbol{W}_K = \boldsymbol{W}_V = \mathbf{I}$, and $\tau = 1/\sqrt{D_{qk}}$. Conversely, the conventional SA mechanism can be viewed as a generalized *mean-shift*:

$$\hat{\boldsymbol{Z}} = \text{SA}(\boldsymbol{Z}) = \mathbf{W}_V \boldsymbol{Z} \,\text{softmax}\left(1/\sqrt{D_{qk}}\boldsymbol{Z}^\top \left(\mathbf{W}_Q^\top \mathbf{W}_K\right) \boldsymbol{Z}\right), \quad (5)$$

with learnable distance measure $\boldsymbol{Z}^\top(\mathbf{W}_Q^\top \mathbf{W}_K)\boldsymbol{Z}$ and projection $\mathbf{W}_V$. Unlike GMM and *k-means*, *mean-shift* is capable of modeling clusters of complex non-convex shape with cluster number automatically determined by local scale (proscribed by covariance) [33]. Hence, it is well-aligned with the semantics of natural images.

**ViT from the perspective of mean-shift** In ViT [22], images are initially tokenized and then processed through a sequence of transformer layers. Each transformer layer is comprised of a skip-connected multi-head SA (MHSA) and a skip-connected MLP. MHSA can be constructed from Eq. 5 with $m$ projections in parallel, *i.e.*, $[\mathbf{W}_Q^h, \mathbf{W}_K^h, \mathbf{W}_V^h], h = 1, \cdots, m$. The $m$ returned modes are then concatenated along channel dimension and reprojected to a single return through

$$\hat{\boldsymbol{Z}} = \text{MHSA}(\boldsymbol{Z}) = \mathbf{W}_O \text{concat}([[\hat{\boldsymbol{Z}}^1], \ldots, [\hat{\boldsymbol{Z}}^m]]) + \mathbf{b}_O. \quad (6)$$

Note that the $\ell_2$ normalization assumed in Eq. 4 is moderately relaxed via layer normalization (LN) to incorporate the extra degree of freedom in the vector magnitude. With skip connection and the one-step *mean-shift* update described in Eqs. 5, 6, a transformer layer essentially finds the local centroid for each query $\boldsymbol{z}$ and drives them closer to the re-projected centroids through $\boldsymbol{z} = \boldsymbol{z} + \hat{\boldsymbol{z}}$, followed by an MLP processing step with skip connection. ViT iterates the process multiple times (*e.g.*, 12 or 24 layers) to capture the contextual and semantic information of an image.

The clustering process above concords with one inductive bias of the attention mechanism represented by the *sparse variable creation* [24], *i.e.*, an SA head learns a sparse function that only depends on a small subset of input coordinates. In the context of clustering, the subset of input corresponds to the modes of density $p(\boldsymbol{z})$. As the high-level semantic information is typically spatially sparse (*e.g.*, the representaion for a RoI in object detection, a single label for a region in segmentation, or a scene-graph, etc.), it is natural to leverage transformer for joint embedding and clustering to learn semantically meaningful representations.

## 4   Methodology

FLSL features a bi-level clustering process (Figure 1), which is formally described as follows.

Given a dataset $\mathcal{X}$ (e.g., a set of images), FLSL learns an encoding scheme $f_\theta : \mathcal{X} \to \mathcal{Z}, \forall \boldsymbol{X} \in \mathcal{X}, \boldsymbol{Z} = f_\theta(\boldsymbol{X})$. $\boldsymbol{Z}$ can be formulated as $\boldsymbol{Z} = \bigcup_c^{N_c} \tilde{\boldsymbol{z}}^c$, where $\tilde{\boldsymbol{z}}^c$ is a subset of $\boldsymbol{Z}$ forming a cluster, $N_c$ is the number of clusters determined by a clustering scheme, *e.g.*, *mean-shift*, and $N_c \leq |\boldsymbol{Z}|$. FLSL aims to encourage the following properties:
(i) **Intra-view**: encodings corresponding to a semantic concept (as a cluster), $\boldsymbol{z} \in \tilde{\boldsymbol{z}}^c$, are close to the cluster representative (*e.g.*, mode) $\hat{\boldsymbol{z}}^c$ and far away from the encodings of other clusters;
(ii) **Inter-view**: the cluster representatives $\hat{\boldsymbol{z}}$s of the positive regions in $\boldsymbol{X}$s over $\mathcal{X}$ are pushed closer to each other.

The FLSL-extracted features should be well-aligned with dense prediction tasks, such as object detection, where the representation of an object or stuff (*i.e.*, cluster of features) are desired to be (i) well-separated from others in an image (locally semantic), and (ii) close to its positive samples in the dataset (globally semantic). In this section, we present the objectives for both levels of clustering, which are then combined to form the final objective.

Figure 2: Overview of the FLSL framework. Similar to DINO [10], FLSL is comprised of a teacher network and a student network, which have the same architecture – a ViT encoder $f$ and a projection head $g$ – but with different parameters. Two *mean-shift* operations: a non-parametric self-attention (SA) and a non-parametric cross-attention (CA) are applied to the last layer of $f_t$, $f_s$ before $g_t$, $g_s$, respectively, and the CA takes output of $f_s$ as queries. The two networks are trained to maximize the agreement between the probability distributions $\boldsymbol{p}_i$s and $\boldsymbol{p}_i^+$s and the agreement between features $\boldsymbol{z}_i$s and their cluster representatives $\hat{\boldsymbol{z}}_i$s.

## 4.1 Intra-view clustering with mean-shift

As discussed in Sec. 3, local semantics of an image can be captured by non-parametric clustering such as *mean-shift*. Hence, with *mean-shift* update rule Eq. 4, it can be proved that the probability of $\boldsymbol{z}_j$ given point $\boldsymbol{z}_i$, $p(\boldsymbol{z}_j|\boldsymbol{z}_i) = [\text{softmax}(\tau \boldsymbol{z}_i^\top \boldsymbol{Z})]_j$, should satisfy:

$$p(\boldsymbol{z}_j|\boldsymbol{z}_i) \geq 1 \Big/ \Big( \Big( \sum_{k \in c_i} e^{(\boldsymbol{z}_i^\top \boldsymbol{z}_k - \boldsymbol{z}_i^\top \boldsymbol{z}_j)\tau} \Big) + (N - |c_i|)e^{-\Delta_{ij}\tau} \Big), \forall j \in c_i \tag{7}$$

where $N = |\boldsymbol{Z}|$, $c_i$ is the set of indices of points in the same cluster including $\boldsymbol{z}_i$, and $\Delta_{ij}$ is the degree of separability defined as $\Delta_{ij} = \boldsymbol{z}_i^\top \boldsymbol{z}_j - \max_{k \in [N] \setminus c_i} \boldsymbol{z}_i^\top \boldsymbol{z}_k$, such that larger $\Delta_{c_i} = \sum_{j \in c_i} \Delta_{ij}$ indicates better separation. For locally semantic encodings, we desire the in-cluster points to be close to each other, or equivalently, to be close to its cluster representative, and stay far away from the out-cluster points, which indicates a large $\Delta$ value. As $\Delta$ becomes sufficiently large, the RHS of Eq. 7 can be approximated as $1/\sum_{k \in c_i} \exp\big((\boldsymbol{z}_i^\top \boldsymbol{z}_k - \boldsymbol{z}_i^\top \boldsymbol{z}_j)\tau\big)$, and for out-cluster points, the probability $p(\boldsymbol{z}_{j \notin c_i}|\boldsymbol{z}_i)$ approaches to 0. This results in a semantics-aligned cluster representative via *mean-shift* – a weighted sum of **only** in-cluster points. This can be realized by contrasting among points using attention map as soft cluster mask to drive the query point $\boldsymbol{z}_i$ closer to the returned mode $\hat{\boldsymbol{z}}_i$. It leads to the **intra-view** clustering objective:

$$\min_{f_\theta} \sum_{i=1}^{N} \|\boldsymbol{z}_i - \hat{\boldsymbol{z}}_i\|_2^2. \tag{8}$$

Proof of Eq. 7 and detailed explanation is provided in Appendix A.

## 4.2 Inter-view clustering with k-means

To learn globally semantic representations, similar to the existing SSL methods, we formulate the problem as a variant of *k-means* clustering. For $\hat{\boldsymbol{z}}$s extracted from an entire dataset, the *k-means* objective with generalized non-empty cluster constraint [4] can be expressed as

$$\min_{\mathcal{M}} \frac{1}{N'} \sum_{\hat{\boldsymbol{z}} \in \hat{\mathcal{Z}}} \sum_{k=1}^{K} \delta_{kk(\hat{\boldsymbol{z}})} \|\hat{\boldsymbol{z}} - \boldsymbol{\mu}_{k(\hat{\boldsymbol{z}})}\|_2^2 + D_{\text{KL}}\left(\bar{\boldsymbol{p}}\|\boldsymbol{\pi}\right), \tag{9}$$

where $\mathcal{M}$ is a set of $K$ centroids $\{\boldsymbol{\mu}_1, \cdots, \boldsymbol{\mu}_K\}$, $\hat{\mathcal{Z}}$ is a set of cluster representatives over the entire dataset, $N' = |\hat{\mathcal{Z}}|$, $k(\hat{\boldsymbol{z}}) = \arg\min_k \|\boldsymbol{\mu}_k - \hat{\boldsymbol{z}}\|_2$, $\delta_{ij}$ is the *Kronecker delta*, with $\delta_{ij} = 1$ iff $i = j$, and 0 otherwise, $[\bar{\boldsymbol{p}}]_{[i]} = 1/N' \sum_{\hat{\boldsymbol{z}}} \delta_{ik(\hat{\boldsymbol{z}})}$, and $\boldsymbol{\pi}$ is the prior, *e.g.*, a vector of the preset proportion for each cluster. With positive pairs $(\hat{\boldsymbol{z}}^+, \hat{\boldsymbol{z}})$ created via data augmentation, the objective can then be constructed as *k-means* clustering with extra separation margin for $\hat{\boldsymbol{z}}^+$:

$$\min_{\mathcal{M}} \frac{1}{N'} \sum_{\hat{\boldsymbol{z}} \in \hat{\mathcal{Z}}} \Bigg( \sum_{k=1}^{K} \delta_{kk(\hat{\boldsymbol{z}})} \|\hat{\boldsymbol{z}} - \boldsymbol{\mu}_{k(\hat{\boldsymbol{z}})}\|_2^2 + \big(1 - \delta_{k(\hat{\boldsymbol{z}}^+)k(\hat{\boldsymbol{z}})}\big) \|\hat{\boldsymbol{z}}^+ - \boldsymbol{\mu}_{k(\hat{\boldsymbol{z}})}\|_2^2 \Bigg) + D_{\text{KL}}\left(\bar{\boldsymbol{p}}\|\boldsymbol{\pi}\right). \tag{10}$$

A common approach to tackle the optimization problem above is to relax the hard cluster assignment constraint $\delta_{ij} \in \{0, 1\}$ to $[0, 1]$ via a **classification head** to $\hat{\boldsymbol{z}}$ with a small temperature ($\ll 1$). This relaxes Eq. 9 to a more general Gaussian Mixture Model (GMM) formulation (cf. Appendix B).

By rewriting $1 - \delta_{k(\boldsymbol{z}^+)k(\boldsymbol{z})}$ in Eq. 10 as $\sum_{k=1}^{K} \delta_{kk(\boldsymbol{z}^+)} - \delta_{kk(\boldsymbol{z}^+)}\delta_{kk(\boldsymbol{z})}$, and with the relaxed hard cluster assignment via a classification head, the objective for the **inter-view** clustering can be

expressed by

$$\min_{\mathcal{M}} \frac{1}{N'} \sum_{\hat{z} \in \hat{z}} \mathrm{H}(\boldsymbol{p}(\hat{\boldsymbol{z}}^+), \boldsymbol{p}(\hat{\boldsymbol{z}})) + D_{\mathrm{KL}}\left(\bar{\boldsymbol{p}} \| \boldsymbol{\pi}\right), \tag{11}$$

where $\boldsymbol{p}(\boldsymbol{x}) = \mathrm{softmax}\left(\tau' \boldsymbol{W}_C^\top \boldsymbol{x}\right)$, $\tau' \ll 1$ with $\boldsymbol{W}_C$ defined as a matrix of $K$ orderly concatenated centroids, and $\mathrm{H}(x, y) = -x \log y$ (cf. Appendix C).

**Positive sample retrieval** Unlike the common instance-level SSL, the positive samples in FLSL are amorphous clusters of features, $(\tilde{\boldsymbol{z}}^+, \tilde{\boldsymbol{z}})$, corresponding to the same semantic concept in two views. In contrast to previous works assigning the best-matching patch [38, 63] or thresholded vicinity [73], we leverage the cluster assignment mechanism inherent in *mean-shift*, where a query $\boldsymbol{z}$ is automatically assigned to a cluster represented by the return $\hat{\boldsymbol{z}}$. For query from another view, the *mean-shift* naturally manifests as a cross-attention (CA),

$$\hat{\boldsymbol{z}}^+ = \boldsymbol{Z}^+ \, \mathrm{softmax}\left(\tau \boldsymbol{z}^\top \boldsymbol{Z}^+\right), \tag{12}$$

With representations semantically coherent on local and global levels, the returned $\hat{\boldsymbol{z}}^+$ from the augmented view $\boldsymbol{Z}^+$ by query $\boldsymbol{z}$ should agree with the returned $\hat{\boldsymbol{z}}$ from the original view. To help establish this semantic constraint, representations at the projected positions from the augmented view can be used as positive samples at the early stage of training. This process can be viewed as data retrieval in dense associative memory recognized in [48].

### 4.3 FLSL Objective

By combining the objectives from the two clustering levels, we arrive at the objective of FLSL:

$$\min \frac{1}{N'} \sum_{\boldsymbol{Z} \in \mathcal{Z}} \sum_{\boldsymbol{z} \in \boldsymbol{Z}} \upsilon \|\boldsymbol{z} - \hat{\boldsymbol{z}}\|_2^2 + \eta \sum_{\boldsymbol{z} \in \boldsymbol{Z}} \mathrm{H}(\boldsymbol{p}(\hat{\boldsymbol{z}}^+), \boldsymbol{p}(\hat{\boldsymbol{z}})) + \gamma D_{\mathrm{KL}}\left(\bar{\boldsymbol{p}} \| \boldsymbol{\pi}\right), \tag{13}$$

$$\text{with } \hat{\boldsymbol{z}} = \mathrm{SA}(\boldsymbol{z}, \boldsymbol{Z}, \boldsymbol{Z}), \ \hat{\boldsymbol{z}}^+ = \mathrm{CA}(\boldsymbol{z}, \boldsymbol{Z}^+, \boldsymbol{Z}^+),$$

where $\upsilon$, $\eta$ and $\gamma$ are the hyperparameters controlling the importance of each term, and the SA and CA above are non-parametric.

Figure 2 illustrates the FLSL framework. We follow the common joint-embedding strategy of SSL, except that we simultaneously maximize the agreement between positive cluster representatives $(\boldsymbol{p}(\hat{\boldsymbol{z}}^+), \boldsymbol{p}(\hat{\boldsymbol{z}}))$ and the agreement between an in-cluster point and its cluster representative $(\boldsymbol{z}, \hat{\boldsymbol{z}})$. The KL-divergence term in Eq. 13 serves as a volume maximization regularizer. Experiments show that the FLSL objective effectively promote locally and globally semantic representations, resulting in significantly improved transferability of learned features to object detection and segmentation. Note that FLSL does not involve a class token in its objective (Eq. 13).

## 5 Experiments

In this section, we evaluate the performance of FLSL by conducting extensive experiments. Specifically, we compare FLSL to existing SSL approaches on multiple dense prediction benchmarks: (i) MS-COCO [42] object detection and instance segmentation, (ii) UAVDT [23] object detection from UAV platforms, and (iii) DAVIS video instance segmentation [46]. Moreover, we investigate the properties of FLSL features in terms of semantic alignment and feature separability in the embedding space. Detailed experimental setups are provided in the respective subsections and supplementary materials. All our experiments are performed on Nvidia RTX A6000.

**Implementation details** The implementation of ViT in our experiments mostly follows DeiT [54] excluding the [class] token. The configuration of the ViT variants utilized in this paper is summarized in Appendix E.3. The coefficients of Eq. 13 in our experiments are $\upsilon = .03$, $\eta = 1$ and $\gamma = 5$ unless stated otherwise. We assume a uniform prior, *i.e.*, $\pi_k = 1/K$, $\forall k$. Models are pretrained on ImageNet-1k [52] dataset using AdamW optimizer [45] with a batch size of 512. We follow the data augmentation from BYOL [27] (*e.g.*, color jittering of brightness, contrast, saturation and hue, Gaussian blur and solarization) with preceding random crops and resizing (to $224 \times 224$) and make them asymmetric. Computation among dense features can be expensive. Therefore, we apply a grid random sampling to the queries. All ViT models are pretrained for 300 epochs as in most baselines for a fair comparison. Pseudo-code, training details, and settings of augmentation pipeline are provided in Appendix E.

| Pretrain | Backbone | Epoch | #Params | $AP^{bbox}$ | $AP^{bbox}_{50}$ | $AP^{bbox}_{75}$ | $AP^{mk}$ | $AP^{mk}_{50}$ | $AP^{mk}_{70}$ |
|---|---|---|---|---|---|---|---|---|---|
| MoCo-v2 | RN50 | 200 | 23M | 38.9 | 59.2 | 42.4 | 35.5 | 56.2 | 37.8 |
| DetCo | RN50 | 200 | 23M | 40.1 | 61.0 | 43.9 | 36.4 | 58.0 | 38.9 |
| DenseCL | RN50 | 200 | 23M | 40.3 | 59.9 | 44.3 | 36.4 | 57.0 | 39.2 |
| BYOL | RN50 | 1000 | 23M | 40.4 | 61.6 | 44.1 | 37.2 | 58.8 | 39.8 |
| SCRL | RN50 | 1000 | 23M | 41.3 | 62.4 | 45.0 | 37.7 | 59.6 | 40.7 |
| MOCO-v3 | ViT-S/16 | 300 | 21M | 39.8 | 62.6 | 43.1 | 37.1 | 59.6 | 39.2 |
| MoBY | ViT-S/16 | 300 | 21M | 41.1 | 63.7 | 44.8 | 37.6 | 60.3 | 39.8 |
| DINO | ViT-S/16 | 300 | 21M | 40.8 | 63.4 | 44.2 | 37.3 | 59.9 | 39.5 |
| DINO+SelfPatch | ViT-S/16 | 200 | 21M | 42.1 | 64.9 | 46.1 | 38.5 | 61.3 | 40.8 |
| ADCLR | ViT-S/16 | 300 | 21M | 44.3 | 65.4 | 47.6 | 39.7 | 62.1 | 41.5 |
| FLSL | ViT-S/16 | 300 | 21M | 44.9 | 66.1 | 48.1 | 40.8 | 64.7 | 44.2 |
| FLSL | ViT-S/8 | 300 | 21M | 46.5 | 69.0 | 51.3 | 42.1 | 65.3 | 45.0 |

Table 1: MASK R-CNN ON COCO

| Pretrain | $AP^{bbox}$ | $AP^{bbox}_{s}$ | $AP^{bbox}_{m}$ | $AP^{bbox}_{l}$ | $AP^{mk}$ |
|---|---|---|---|---|---|
| None | 48.1 | - | - | - | 42.6 |
| IN-1k Supv. | 47.6 | - | - | - | 42.4 |
| IN-21k Supv. | 47.8 | - | - | - | 42.6 |
| IN-1k DINO | 48.9 | 32.9 | 52.2 | 62.4 | 43.7 |
| IN-1k MAE | 51.2 | 34.9 | 54.7 | 66.0 | 45.5 |
| IN-1k FLSL | 53.1 | 36.9 | 56.2 | 67.4 | 47.0 |

Table 2: VITDET-B/16 WITH MASK R-CNN ON COCO

| Pretrain | Backbone | $AP_{VOC}$ |
|---|---|---|
| IN-1k DINO | ViT-S/16 | 48.9 |
| IN-1k DINO | ViT-B/16 | 49.1 |
| IN-1k DINO | ViT-S/8 | 51.1 |
| IN-1k FLSL | ViT-S/16 | 53.1 |
| IN-1k FLSL | ViT-B/16 | 53.5 |
| IN-1k FLSL | ViT-S/8 | 55.2 |

Table 3: FASTER R-CNN FPN ON UAVDT

**Baselines** We compare FLSL with various existing SSL approaches that are based on the ResNet [31] and ViT [22] architectures: (a) self-supervised ResNet: MoCo-v2 [15], DetCo [69], DenseCL [63], BYOL [27], and SCRL [50]; and (b) self-supervised ViT: MoCo-v3 [16], MoBY [72], DINO [10], MAE [28], SelfPatch [75], and ADCLR [76].

**Protocol for hyperparameter tuning** Standard instance-level SSL evaluation protocols typically utilize one of the two approaches: employing a $k$-NN classifier or training a linear classifier on fixed features. Since FLSL learns dense semantic representations rather than a single instance-level representation, both standard evaluation protocols are not suitable for evaluating FLSL in training. Moreover, fine-tuning on a downstream dense prediction tasks can be computationally expensive due to complex prediction heads, and may introduce task-specific biases during hyperparameter tuning. Therefore, we design a bbox-aligned $k$-NN classifier modified from [67] to evaluate the feature quality directly without additional network tuning. Here is an overview of the method. Features of the training data are first extracted with a fixed model. These features are then aligned with their corresponding bounding boxes provided by ILSVRC [51]. For each image, a certain number of representative features $\hat{z}$s (*e.g.*, 9) are selected by a partition criterion and stored in memory. The $k$-NN classifier matches each selected features to its $k$-nearest stored features, which collectively vote for its label. A feature is considered successfully classified if any of the representative features match its class. This protocol is employed for hyperparameter tuning and ablation study of the FLSL pipeline. Appendix F provides further details on the choice of $k$, implementation specifics and evaluation results.

## 5.1 MS-COCO Object Detection & Segmentation

We adopt Mask R-CNN detection framework by incorporating three variants of ViT: (i) ViT-S/16 with FPN [41], (ii) ViT-S/8 with FPN, and (iii) ViT-B/16 with simple feature pyramid (ViTDet) [40]. Models of (i) and (ii) are fine-tuned following the multi-scale training [66, 6] under the standard 1× schedule for a fair comparison. For the model of (iii), we follow the training recipe of [40] and fine-tune the model for 100 epochs.

**Results**. Table 1 reports the detection and segmentation performance of ViT-S/16 and ViT-S/8 with Mask R-CNN [30] on COCO. Specifically, FLSL with ViT-S/16 outperforms ADCLR [76] by +0.6% and +1.1%, and substantially outperforms DINO+SelfPatch [75] by +2.8% and +2.4% on detection ($AP^{bbox}$) and segmentation ($AP^{mk}$), respectively. Both baseline methods feature patch-level contrastive learning. Unlike SelfPatch contrasting between patches within the adjacent neighborhood and ADCLR contrasting via learned queries of random crops, FLSL contrasts the representatives (modes) of feature clusters, which aligns closer with the downstream tasks and thus leads to superior performance. Notably, FLSL with ViT-S/8 further improves the performance by a large margin of +4.4% in $AP^{bbox}$ and +3.6% $AP^{mk}$ over SelfPatch. Table 2 summarizes the results of ViTDet. FLSL shows large performance gains over the DINO baseline by +4.2% $AP^{bbox}$ and +3.3% $AP^{mk}$. FLSL also outperforms the SOTA generative approach, MAE, by +1.7% and +1.4% in the two tasks, respectively.

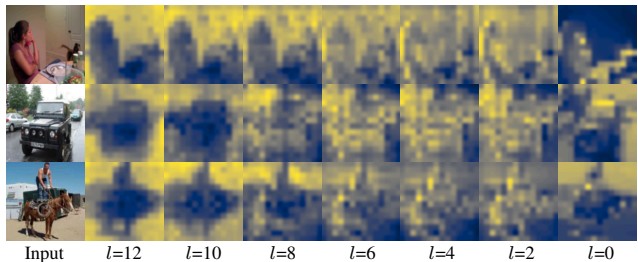

| Pretrain | Arch. | $(\mathcal{J}\&\mathcal{F})_m$ | $\mathcal{J}_m$ | $\mathcal{F}_m$ |
|---|---|---|---|---|
| IN-1k supv. | ViT-S/8 | 66.0 | 63.9 | 68.1 |
| VLOG CT | RN50 | 48.7 | 46.4 | 50.0 |
| YT-VOS MAST | RN18 | 65.5 | 63.3 | 67.6 |
| IN-1k DINO | ViT-S/16 | 61.8 | 60.2 | 63.4 |
| IN-1k DINO | ViT-B/16 | 62.3 | 60.7 | 63.9 |
| IN-1k DINO | ViT-S/8 | 69.9 | 66.6 | 73.1 |
| IN-1k FLSL | ViT-S/16 | 65.6 | 62.4 | 69.4 |
| IN-1k FLSL | ViT-B/16 | 66.1 | 62.9 | 70.0 |
| IN-1k FLSL | ViT-S/8 | 73.5 | 69.9 | 78.1 |

Figure 3: Visualization of the maps of the *aggregated attention score* (ASS) from different layers of ViT-S/16. $l = 0$ denotes the projection layer. As layer goes deeper, the map becomes more partitioned with brightness aligned with the area of the underlying semantic region, *e.g.*, objects or stuff.

Table 4: DAVIS 2017 VIDEO INSTANCE SEGMENTATION. We evaluate the quality of frozen features on video instance tracking. We report mean region similarity $\mathcal{J}_m$ and mean contour-based accuracy $\mathcal{F}_m$.

## 5.2 Small Object Detection: UAVDT

To assess the transferability of FLSL beyond the datasets of common images like COCO, we further investigate its performance on a UAV benchmark, UAVDT [23], which exhibits significant domain shifts from common images (*i.e.*, images captured by ground-level cameras). We utilize Faster R-CNN framework [49] with the same ViT variants used in the COCO experiments and follow the training settings outlined in ClusDet [74]. All ViT-backboned models are trained with $1\times$ schedule.

**Result** Table 3 presents the performance of ViT-S/16, ViT-S/8, and ViT-B/16 with Faster R-CNN for detection tasks on UAVDT under different pretrain schemes. We utilize the official evaluation method in [23], which calculates the class-agnostic VOC AP exclusive of the predictions that falls in the ignored areas. FLSL consistently outperforms DINO (a typical instance-level SSL for ViT) across all three ViT variants by a significant margin. With smaller objects and an imbalanced foreground-background ratio, the significance of local semantics becomes evident. Models require local context to discover small objects and make accurate predictions rather than relying solely on the global semantics of the entire image. This situation aligns well with the strengths of FLSL.

## 5.3 DAVIS Segmentation

To further assess the quality of frozen features learned by FLSL, we evaluate FLSL-pretrained ViT models on DAVIS2017 [46], following the evaluation protocol in [36, 10] that requires fixed representations with no extra training.

**Results** Table 4 shows that FLSL consistently outperforms DINO across all ViT variants in our experiments. The protocol evaluates the quality of learned dense features via segmenting scenes with $k$-nearest neighbors ($k = 5$) within a fixed window ($12 \times 12$) between consecutive frames. This requires dense features to be locally semantic, *i.e.*, features corresponding to the same semantics should be more correlated. Therefore, the improved performance confirms that FLSL encourages model to extract locally semantic representations.

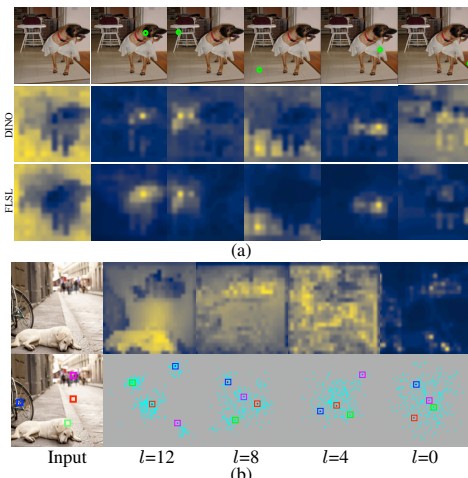

Figure 4: (a) visualization of attention probing by query patches (marked out in green circle in the top row) from the last layer of ViT-S/16 pretrained with FLSL and with DINO. FLSL encourages the model to learn semantic correlations among patches; (b) visualization of separability of the dense representations throughout the transformer (ViT-S/16).

## 5.4 Alignment with Image Semantics

To qualitatively show that FLSL is better aligned with the semantic layout of an image than the common SSL methods, Figure 4(a) compares the self-attention probing maps for features learned via FLSL and DINO. Features from the last layer are used for evaluation. The visualizations are obtained with $224^2$ images. Positions of the query tokens are marked out in green circle in the top row. As shown in the middle and bottom rows of the figure, DINO promotes more correlated attention (*i.e.*, less separation between tokens of query-related area and that of the rest image), while FLSL encourages

| Sinkhorn | $\eta$ | $\gamma$ | $v=0.0$ | $v=.01$ | $v=.02$ | $v=.03$ | $\sim$ | $v=.1$ |
|---|---|---|---|---|---|---|---|---|
| ✓ | 1.0 | 1.0 | 0.1 | 68.7 | 70.7 | 71.2 | $\sim$ | 65.1 |
| × | 1.0 | 1.0 | - | - | - | 66.6 | - | - |
| × | 1.0 | 5.0 | - | - | - | 72.4 | - | - |

Table 5: IMPACT OF COEFFICIENTS IN THE FLSL OBJECTIVE.

| $K$ | 1024 | 2048 | 4096 | 8192 | 16384 |
|---|---|---|---|---|---|
| $k$-NN top-1 | 68.1 | 72.1 | 72.4 | 72.5 | 72.1 |

Table 6: IMPACT OF NUMBER OF CENTROIDS $K$

attention to the regions of high semantic relevance with the query tokens and results in clearer maps consistent with the underlying objects/stuff.

## 5.5 Feature Distribution and Separability

We demonstrate the qualitative results by visualizing the Aggregated Similarity Score (ASS) and the feature distribution in the embedding space using t-sne [57] in Figure 3 and Figure 4(b), respectively. To generate the map of ASS, we sum up the cosine-similarity maps of all tokens, normalize the resulting map with its maximum score and visualize it as a thermal image, *i.e.*, the brighter the pixel, the higher the score. For a semantically well-separated image, each patch only attends to the patches of its own semantic region, *e.g.*, a patch of an object has high similarity scores only with the patches of that object and low scores with the rest. This results in an image with partitions of different brightness proportional to the area of that region, *i.e.*, ideally the larger the size of an object/stuff, the brighter the color. As shown in Figure 3, as the layer goes deeper, the brightness partition of the ASS is more consistent with the underlying objects and stuff in the images (*e.g.*, person, vehicles, horse, switches, wall, and ground, etc.), which indicates the desired separation of the learned features. This is also reflected in the t-sne visualization of the embeddings in Figure 4(b), where the representations become more clustered and separated as the attention layer goes deeper.

## 5.6 Ablation Study

Due to limited space, we present two major ablation studies in this section to help understand the effectiveness of FLSL. The model considered for this entire study is ViT-S trained with 100 epochs. We refer the reader to Appendix I for the complete work.

**Impact of coefficients in the FLSL objective** The FLSL objective (Eq. 13) contains three components: (1) similarity between $\ell_2$-normalized $z$ (features) and $\hat{z}$ (modes), (2) cross-entropy of the probabilities of an augmented pair $H(p(\hat{z}^+), p(\hat{z}))$, and (3) the volume maximization regularizor $D_{\text{KL}}(\bar{p}\|\pi)$. It is computationally expensive to optimally determine the values of more than two coefficients by performing grid search, especially when the ratios among them are large. We tackle this problem by first fixing $\eta = 1$ and setting $\gamma = 1$ along with Sinkhorn normalization [19] to perform a grid search on the value of $v$ with the empirical base condition $v \leq 1$ and $\gamma \geq 1$ [75, 1]. With the fixed $v$, we then perform another grid search on $\gamma$ without Sinkhorn normalization. We implement Sinkhorn normalization as the softmax operation along the batch dimension. Table 5 summarizes the score of bbox-aligned $k$-NN evaluation using different coefficient settings.

**Impact of number of centroids $K$** FLSL is formulated as an explicit clustering problem, with the output dimension of the last fully-connected layer equal to the number of centroids $K$. Compared to its instance-level counterpart DINO [10], FLSL enjoys a smaller output dimension (shown in Table 6). This is because images have higher feature variance compared to feature clusters. For example, an image in ImageNet may contain diverse content from different categories, requiring a large number of centroids to cover the distribution. In contrast, a semantic cluster contains highly correlated features, such as similar textures or objects from the same category, thus requiring fewer centroids. Experimentally, we find that a large number of centroids benefits performance, but is detrimental and costly when being too large. We pick $K = 4,096$ for all our experiments as it strikes a good balance between performance and cost-effectiveness.

More experiment results on semantic segmentation and ablations including the impact of batch size and random pooling window size are relegated to Appendix.

# 6 Conclusions

This paper proposes FLSL, a feature-level self-supervised learning method that bridges the gap between the current SSL methods and downstream dense prediction tasks. We demonstrate for the first time the underlying *mean-shift* clustering process of ViT, which aligns well with natural image semantics. Facilitated by ViT for joint embedding and feature clustering, FLSL performs a bi-level clustering: (i) intra-view clustering to extract the representatives for clusters of features within an image, and (ii) inter-view clustering to encourage the representatives to be globally semantic over

the entire dataset. FLSL achieves a significant improvement over the SOTAs in the dense prediction tasks, including object detection and instance segmentation.

**Limitations and broader impacts** FLSL does not have any significant limitations other than the method is more complex (due to its bi-level clustering) than other SSL methods, and it currently only fits for ViT-based models on dense prediction tasks. Exploring ways to extend FLSL for tasks that necessitate a global representation while retaining its existing properties could be a potential future work. As far as we can foresee, there is no negative societal impact.

## 7 Acknowledgment

This research was sponsored by the Army Research Laboratory under Cooperative Agreement #W911NF-22-2-0025. The views and conclusions contained in this document are those of the authors and should not be interpreted as representing the official policies, either expressed or implied, of the Army Research Laboratory or the U.S. Government. The U.S. Government is authorized to reproduce and distribute reprints for Government purposes notwithstanding any copyright notation herein.

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
