# FLSL: Feature-level Self-supervised Learning

*Supplementary Materials*

## A  Intra-veiw clustering with mean-shift

An image can be represented as an empirical probability density function that comprises amorphous clusters of features. Given a dense representation of an image $\boldsymbol{Z} = \{\boldsymbol{z}_i\}_{i=1}^N$ and the *mean-shift* clustering scheme, the conditional probability of $\boldsymbol{z}_j$ given $\boldsymbol{z}_i$ indicates the probability of feature $\boldsymbol{z}_i$ being assigned to the cluster of $\boldsymbol{z}_j$, which is defined as follows:

$$p(\boldsymbol{z}_j|\boldsymbol{z}_i) = \left[\mathrm{softmax}\left(\tau \boldsymbol{z}_i^\top \boldsymbol{Z}\right)\right]_j \tag{14}$$

$$= e^{\tau \boldsymbol{z}_i^\top \boldsymbol{z}_j} \bigg/ \left( \sum_{k \in c_i} e^{\tau \boldsymbol{z}_i^\top \boldsymbol{z}_k} + \sum_{k \in [N] \setminus c_i} e^{\tau \boldsymbol{z}_i^\top \boldsymbol{z}_k} \right)$$

$$= 1 \bigg/ \left( \left( \sum_{k \in c_i} e^{-(\boldsymbol{z}_i^\top \boldsymbol{z}_j - \boldsymbol{z}_i^\top \boldsymbol{z}_k)\tau} \right) + \sum_{k \in [N] \setminus c_i} e^{-(\boldsymbol{z}_i^\top \boldsymbol{z}_j - \boldsymbol{z}_i^\top \boldsymbol{z}_k)\tau} \right)$$

$$\geq 1 \bigg/ \left( \left( \sum_{k \in c_i} e^{-(\boldsymbol{z}_i^\top \boldsymbol{z}_j - \boldsymbol{z}_i^\top \boldsymbol{z}_k)\tau} \right) + (N - |c_i|)e^{-\Delta_{ij}\tau} \right), \tag{15}$$

where $\tau$ is the inverse temperature, $c_i$ is the set of indices of points contained in the cluster of $\boldsymbol{z}_i$, $[N] = \{1, \ldots, N\}$, and $\Delta_{ij}$ is the cluster separation with respect to $\boldsymbol{z}_i$, defined as

$$\Delta_{ij} = \boldsymbol{z}_i^\top \boldsymbol{z}_j - \max_{m \in [N] \setminus c_i} \boldsymbol{z}_i^\top \boldsymbol{z}_m, \; j \in c_i, \tag{16}$$

measuring the gain of similarity between $\boldsymbol{z}_i$ and an in-cluster point $\boldsymbol{z}_j$ over the similarity between $\boldsymbol{z}_i$ and the out-cluster point $\boldsymbol{z}_k$ that is closest to $\boldsymbol{z}_i$.

To achieve locally semantic representations, our objective is for the points within each cluster to be in close proximity to each other or, equivalently, close to their cluster representative. This proximity ensures consistency in encoded semantics. Additionally, we aim for these in-cluster points to be distinctly separated from the points outside the cluster. This separation encourages well-defined clusters to accurately reflect different semantics, i.e., a large $\Delta_{ij}$ and a small in-cluster variance. As $\Delta$ becomes sufficiently large (with a proper inverse temperature), the RHS of Eq. 15 can be approximated as $1/\sum_{k \in c_i} e^{-(\boldsymbol{z}_i^\top \boldsymbol{z}_j - \boldsymbol{z}_i^\top \boldsymbol{z}_k)\tau}$ for in-cluster points, $i, j \in c_i$. Meanwhile, the posterior for the out-cluster points, $p(\boldsymbol{z}_{j \notin c_i}|\boldsymbol{z}_i)$, approaches 0 at the rate of

$$p(\boldsymbol{z}_{j \notin c_i}|\boldsymbol{z}_i) \leq 1 \bigg/ \left( \left( \sum_{k \in c_i} e^{\tau \min_{k \in c_i} \Delta_{ik}} \right) + (N - |c_i|)e^{-\tau \max_{k \notin c_i}(\boldsymbol{z}_i^\top \boldsymbol{z}_j - \boldsymbol{z}_i^\top \boldsymbol{z}_k)} \right). \tag{17}$$

The resulting return of a single *mean-shift* update becomes

$$\hat{\boldsymbol{z}}_i = \boldsymbol{Z}\mathrm{softmax}\left(\tau \boldsymbol{z}_i^\top \boldsymbol{Z}\right) = \sum_{j \in [N]} p(\boldsymbol{z}_j|\boldsymbol{z}_i)\boldsymbol{z}_j \approx \sum_{j \in c_i} \frac{1}{\sum_{k \in c_i} e^{-(\boldsymbol{z}_i^\top \boldsymbol{z}_j - \boldsymbol{z}_i^\top \boldsymbol{z}_k)\tau}} \boldsymbol{z}_j + \boldsymbol{0}, \tag{18}$$

which is essentially a weighted sum of the in-cluster points **only**. To promote the aforementioned property while maintaining low in-cluster variance, one approach is to drive the point closer to its cluster representative by optimizing

$$\min \sum_{i=1}^N \|\boldsymbol{z}_i - \hat{\boldsymbol{z}}_i\|_2^2, \quad \text{with } \hat{\boldsymbol{z}}_i = \boldsymbol{Z}\mathrm{softmax}(\tau \boldsymbol{z}_i^\top \boldsymbol{Z}). \tag{19}$$

Notably, with a large inverse temperature $\tau \gg 1$, a single *mean-shift* update becomes the single-step pattern retrieval mechanism in dense associative memory (DAM) [37, 48].

# B  The GMM formulation of the constrained k-means objective

The *k-means* objective with generalized non-empty cluster constraint [4] can be expressed as

$$\min_{\mathcal{M}} \frac{1}{N'} \sum_{\hat{z} \in \hat{\mathcal{Z}}} \sum_{k=1}^{K} \delta_{kk(\hat{z})} \|\hat{z} - \boldsymbol{\mu}_{k(\hat{z})}\|_2^2 + D_{\mathrm{KL}}\left(\bar{\boldsymbol{p}} \| \boldsymbol{\pi}\right), \tag{20}$$

where $\mathcal{M}$ is a set of $K$ centroids $\{\boldsymbol{\mu}_1, \cdots, \boldsymbol{\mu}_K\}$, $\hat{\mathcal{Z}}$ is a set of cluster representatives over the entire dataset, $N' = |\hat{\mathcal{Z}}|$, $k(\hat{z}) = \arg\min_k \|\boldsymbol{\mu}_k - \hat{z}\|_2$, $\delta_{ij}$ is the *Kronecker delta*, with $\delta_{ij} = 1$ iff $i = j$, and $0$ otherwise, $[\bar{\boldsymbol{p}}]_{[i]} = 1/N' \sum_{\hat{z}} \delta_{ik(\hat{z})}$, and $\boldsymbol{\pi}$ is the prior, *e.g.*, a vector of the preset proportion for each cluster.

As mentioned in the main paper, a common approach to tackle the optimization problem above is to relax the hard cluster assignment constraint $\delta_{ij} \in \{0, 1\}$ to $[0, 1]$ with a classification head to $\hat{z}$. This relaxes Eq. 20 to the more general Gaussian Mixture Model (GMM) formulation, allowing each point to have a partial membership of each cluster with a certain probability. The GMM ELBO can be expressed by the average term-by-term reconstruction and KL to prior as

$$\mathcal{L}(\theta, \mathcal{M}, \boldsymbol{\Sigma}) = -\frac{1}{N'} \left( \sum_{\hat{z} \in \hat{\mathcal{Z}}} \sum_{\boldsymbol{\mu} \in \mathcal{M}} q(\boldsymbol{\mu}|\hat{z}) d(\hat{z}, \boldsymbol{\mu}; \boldsymbol{\Sigma}_{\boldsymbol{\mu}}) + \sum_{\hat{z} \in \hat{\mathcal{Z}}} D_{\mathrm{KL}}\left(q(\boldsymbol{\mu}|\hat{z}) \| \boldsymbol{\pi}\right) \right) + C, \tag{21}$$

where $d(\boldsymbol{z}, \boldsymbol{\mu}; \boldsymbol{\Sigma}_{\boldsymbol{\mu}}) = (\boldsymbol{z} - \boldsymbol{\mu})^\top \boldsymbol{\Sigma}_{\boldsymbol{\mu}}^{-1} (\boldsymbol{z} - \boldsymbol{\mu})$ is the *Mahalanobis* distance, $C$ is a constant under the assumption of homoscedastic and isotropic Gaussian kernel. With a classification head, the posterior of $\hat{z}$ belonging to cluster $k$ is

$$q(\boldsymbol{\mu}_k|\hat{z}) = \left[\mathrm{softmax}\left(\tau' \boldsymbol{W}_{\mathcal{M}}^\top \hat{z} + \log \boldsymbol{\pi} - \tau' \left(\hat{z}^\top \hat{z} + \mathrm{diag}\left(\boldsymbol{W}_{\mathcal{M}}^\top \boldsymbol{W}_{\mathcal{M}}\right)\right)\right)\right]_k, \tag{22}$$

where $\tau'$ is the inverse temperature, and $\boldsymbol{W}_{\mathcal{M}}$ is a matrix of $K$ concatenated centroids with its $k$th column corresponding to $\boldsymbol{\mu}_k$. Particularly, we assume all vectors are $\ell_2$-normalized. This further simplifies the posterior to $q(\boldsymbol{\mu}|\hat{z}) = \mathrm{softmax}(\tau' \boldsymbol{W}_{\mathcal{M}}^\top \hat{z} + \log \boldsymbol{\pi})$, which conforms with the output of a classification head as a mixing proportion.

The hard cluster assignment in Eq. 20 can be recovered by sharpening the posterior with a small covariance, or equivalently, a large inverse temperature $\tau'$, *i.e.*,

$$\lim_{\tau' \to \infty} q_\phi(\boldsymbol{\mu}_k|\hat{z}) = \lim_{\tau' \to \infty} \left[\mathrm{softmax}(\tau' \boldsymbol{W}_{\mathcal{M}}^\top \hat{z} + \log \boldsymbol{\pi})\right]_k$$
$$= \lim_{\tau' \to \infty} \left[\mathrm{softmax}(\tau' \boldsymbol{W}_{\mathcal{M}}^\top \hat{z})\right]_k = \delta_{kk(\hat{z})}. \tag{23}$$

With a sufficiently large inverse temperature, the KL-divergence term of Eq. 21 becomes

$$\frac{1}{N'} \sum_{\hat{z} \in \hat{\mathcal{Z}}} D_{\mathrm{KL}}\left(\delta_{kk(\hat{z})} \| \boldsymbol{\pi}\right) = -\sum_{k=1}^{K} \frac{N'_k}{N'} \log \boldsymbol{\pi}_k, \tag{24}$$

where $N'_k = \sum_{\hat{z} \in \hat{\mathcal{Z}}} \mathbf{1}_{[\boldsymbol{k}(\hat{z}) = k]}$. By defining $[\bar{\boldsymbol{p}}]_k = \frac{N'_k}{N'}$ and adding back the non-empty constraint as the negative entropy of $\bar{\boldsymbol{p}}$, the resulting GMM ELBO recovers Eq. 9 with $d(\hat{z}, \boldsymbol{\mu}; \boldsymbol{\Sigma}_{\boldsymbol{\mu}}) \propto \|\hat{z} - \boldsymbol{\mu}_{k(\hat{z})}\|_2^2$.

# C  The cross-entropy formulation of the constrained k-means with positive samples

With positive pairs $(\hat{z}^+, \hat{z})$ created via data augmentation, the constrained *k-means* objective in Eq. 20 can be formulated as *k-means* clustering with an extra separation margin for $\hat{z}^+$.

Here, we present the derivation of Eq. 11 in the main paper, considering a more general setting that involves multiple positive samples $\{\hat{z}^{(a)}\}_{a=1}^A$ anchored on $\hat{z}^{(0)} = \hat{z}$ through data augmentation. The objective in Eq. 10 from the main paper is essentially a special case of the following expression, where the number of positive pairs $A$ equal to 1:

$$\min_{\mathcal{M}} \frac{1}{N'} \sum_{\hat{z} \in \hat{\mathcal{Z}}} \left( \sum_{k=1}^{K} \delta_{kk(\hat{z})} \|\hat{z} - \boldsymbol{\mu}_{k(\hat{z})}\|_2^2 + \frac{1}{A} \sum_{a=1}^{A} \left(1 - \delta_{k(\hat{z}^{(a)})k(\hat{z})}\right) \|\hat{z}^{(a)} - \boldsymbol{\mu}_{k(\hat{z})}\|_2^2 \right) + D_{\mathrm{KL}}\left(\bar{\boldsymbol{p}} \| \boldsymbol{\pi}\right), \tag{25}$$

which imposes that a point and its positive samples reside in the same cluster.

The above optimization problem can be tackled by minimizing its upper bound with a relaxed hard assignment. Specifically, the term inside the parenthesis is bounded by

$$\sum_{k=1}^{K} \delta_{kk(\hat{z}^{(0)})} \|\hat{z}^{(0)} - \boldsymbol{\mu}_{k(\hat{z}^0)}\|_2^2 + \frac{1}{A} \sum_{a=1}^{A} \left(1 - \delta_{k(\hat{z}^{(a)})k(\hat{z}^{(0)})}\right) \|\hat{z}^{(a)} - \boldsymbol{\mu}_{k(\hat{z}^{(0)})}\|_2^2$$

$$\leq \|\hat{z}^{(0)} - \boldsymbol{\mu}_{k(\hat{z}^{(0)})}\|_2^2 + \frac{1}{A} \max_{a \in A} \|\hat{z}^{(a)} - \boldsymbol{\mu}_{k(\hat{z}^{(0)})}\|_2^2 \sum_{a=1}^{A} \left(1 - \delta_{k(\hat{z}^{(a)})k(\hat{z}^{(0)})}\right). \quad (26)$$

By rewriting $1 - \delta_{k(\hat{z}^{(a)})k(\hat{z}^{(0)})}$ as $\sum_{k=1}^{K} \left(\delta_{kk(z^{(a)})} - \delta_{kk(\hat{z}^{(a)})}\delta_{kk(\hat{z}^{(0)})}\right)$, the RHS of Eq. 26 becomes

$$\|\hat{z}^{(0)} - \boldsymbol{\mu}_{k(\hat{z}^{(0)})}\|_2^2 + \frac{1}{A} \max_{a \in A} \|\hat{z}^{(a)} - \boldsymbol{\mu}_{k(\hat{z}^{(0)})}\|_2^2 \sum_{a=1}^{A} \left(\sum_{k=1}^{K} \delta_{kk(\hat{z}^{(a)})} - \sum_{k=1}^{K} \delta_{kk(\hat{z}^{(a)})}\delta_{kk(\hat{z}^{(0)})}\right)$$

$$= \|\hat{z}^{(0)} - \boldsymbol{\mu}_{k(\hat{z}^{(0)})}\|_2^2 + \frac{1}{A} \max_{a \in A} \|\hat{z}^{(a)} - \boldsymbol{\mu}_{k(\hat{z}^{(0)})}\|_2^2 \sum_{a=1}^{A} \sum_{k=1}^{K} \left(\delta_{kk(\hat{z}^{(a)})} \left(1 - \delta_{kk(\hat{z}^{(0)})}\right)\right), \quad (27)$$

which is bounded by

$$\leq \|\hat{z}^{(0)} - \boldsymbol{\mu}_{k(\hat{z}^{(0)})}\|_2^2 + \frac{1}{A} \max_{a \in A} \|\hat{z}^{(a)} - \boldsymbol{\mu}_{k(\hat{z}^{(0)})}\|_2^2 \sum_{a=1}^{A} \sum_{k=1}^{K} -\delta_{kk(\hat{z}^{(a)})} \log\left(\delta_{kk(\hat{z}^{(0)})} + \epsilon\right), \quad (28)$$

with $0 < \epsilon \ll 1$.

To our interest, we assume all vectors are $\ell_2$-normalized. Thus, the bound in Eq. 28 can be further simplified to

$$4 + 4\frac{1}{A} \sum_{a=1}^{A} \sum_{k=1}^{K} -\delta_{kk(\hat{z}^{(a)})} \log\left(\delta_{kk(\hat{z}^{(0)})} + \epsilon\right). \quad (29)$$

By relaxing the hard assignment $\delta_{kk(\hat{z})} \in \{0, 1\}$ to $[0, 1]$ using a classification head to $\hat{z}$ as in the GMM formulation in Appendix B with a sufficiently large inverse temperature $\tau' \gg 1$, the optimization in Eq. 25 can be approached by

$$\min_{\mathcal{M}} \frac{1}{AN'} \sum_{a=1}^{A} \sum_{\hat{z} \in \hat{\mathcal{Z}}} \mathrm{H}(\boldsymbol{p}(\hat{z}^{(a)}), \boldsymbol{p}(\hat{z})) + D_{\mathrm{KL}}\left(\bar{\boldsymbol{p}} \| \boldsymbol{\pi}\right), \quad (30)$$

where $\boldsymbol{p}(\hat{z}) = q(\boldsymbol{\mu}|\hat{z}) = \mathrm{softmax}\left(\tau' \boldsymbol{W}_{\mathcal{M}}^\top \hat{z}\right)$, and $\mathrm{H}(\boldsymbol{x}, \boldsymbol{y}) = -\boldsymbol{x}^\top \log \boldsymbol{y}$. When $A = 1$, *i.e.*, only considering a single positive pair, the above objective degenerates to Eq. 11 in the main paper.

| model | #blocks | dim | #heads | #tokens | #params | im/s |
|---|---|---|---|---|---|---|
| ViT-S/16 | 12 | 384 | 6 | 196 | 21M | 1,007 |
| ViT-S/8 | 12 | 384 | 6 | 785 | 21M | 180 |
| ViT-B/16 | 12 | 768 | 12 | 196 | 85M | 312 |

Table 8: ViT CONFIGURATION

# D  Other related works

**Unsupervised learning with grouping**  This is intimately connected to self-supervised learning. Early research employed dimensionality reduction techniques, such as PCA and LDA, in conjunction with clustering algorithms like k-means and spectral clustering. The objective was to enable iterative subspace selection paired with clustering. In recent years, the use of non-linear transformations through deep neural networks has been explored. [56] introduces an autoencoder based on Restricted Boltzmann Machines (RBMs) for t-SNE embedding. Meanwhile, in [71], a deep neural network is employed to concurrently learn cluster centroids and feature embeddings. This ensures that the soft assignments of embeddings, based on the centroids, align with a specific target distribution.

Recent works go beyond nonlinear embedding by jointly optimizing the feature and the cluster assignment. DeepCluster [7, 8] utilize k-means to generate pseudo-class labels and applies supervised learning to iteratively fine-tune the model. Local Aggregation (LA) [79] determines a neighborhood

for each instance via clustering and conducts instance-level discrimination solely within these neighborhoods. Conversly, CLD [64] incorporates local clustering into contrastive metric learning and utilizing a cross-level instance-group discrimination approach. PCL [39] compares instance features with group centroids obtained through global clustering per epoch. Meanwhile, SegSort [34] extends representation learning from classification to segmentation. It achieves this by learning a feature for each pixel, operating under the assumption that pixels within the same region inherently form a cluster in the feature space.

**Unsupervised object discovery** Recent success of unsupervised object discovery is highly relevant to the underlying clustering problem of common SSL. A notable approach involves formulating the clustering problem as an optimization of region proposals, while maximizing the cumulative similarity of these proposed regions over a collection of images [59, 60, 61]. Particularly, rOSD [60] utilizes hierarchical saliency clusters constructed from feature maps to generate region proposals. Features within a thresholded vicinity of a local maximum of the saliency map are grouped together, while LOST [53] localizes the position of the smallest object in an image by finding a group of features with the minimum cumulative similarity. Notably, one critical component for the aforementioned methods to be effective is the feature-level clustering manifested by the topological data clustering algorithm, e.g. *selective search* [55], *persistence* [12] in rOSD, or the feature clusters transferred from an SSL-pretrained model (DINO [10]) in LOST. Still, none of them is involved in training or differentiable, leading to sub-optimal solutions.

**Self-attention as clustering** Concurrently with our research, several studies have demonstrated the connection between attention and the clustering process. For example, [77] interprets the attention mechanism using the lens of the information bottleneck (IB), which is also tied to clustering. The IB formulation presented in this study is essentially an EM-fitting of a GMM with soft assignment, based on key assumptions such as the Gaussian approximation in KL minimization and a minor smoothing scale, as detailed in the appendix. In contrast with soft GMM, mean-shift clustering operates non-parametrically (KDE) and does not impose prior assumptions on the structure of the clusters. For instance, it does not predefine the number of clusters and does not involve KL minimization. This characteristic makes mean-shift clustering more aligned with the attention mechanism, which typically doesn't impose many assumptions on its input.

# E  Implementation details

## E.1  Network configuration

We follow the implementation used in DeiT [54] for all the ViT variants used in our experiments, and their configurations are summarized in Table 8.

In the table, "#blocks" is the number of transformer blocks, "dim" is the channel dimension, "#heads" is the number of heads in multi-head attention, "#tokens" is the length of the token sequence when considering $224^2$ resolution inputs, "#params" is the total number of parameters (without counting the projection head), and "im/s" is the inference speed on a NVIDIA V100 GPU with 128 samples per forward.

## E.2  Training details

The implementation of ViT in our experiments mostly follows DeiT [54], with the exception of excluding the `[class]` token. During pretext training, we set the coefficients in the FLSL objective as follows: $\upsilon = .03$, $\eta = 1.0$, and $\gamma = 5.0$, and assume a uniform prior, *i.e.*, $\pi_k = 1/K$, $\forall k$, with the number of centroids $K = 4096$. We pretrain the models on ImageNet-1k dataset without labels using AdamW optimizer [45] and a batch size of 512. In line with DINO, the learning rate linearly ramps up during the first 10 epochs to the base value determined with the linear scaling rule [26]: $lr = 0.00025$ with the reference *batch_size* $= 256$. The warm-up is followed by the learning rate decay governed by cosine schedule [44] with the target learning rate $10^{-6}$. The weight decay also governed by a cosine schedule from 0.05 to 0.5. The update rule for teacher network is $\theta_t \leftarrow \lambda\theta_t + (1 - \lambda)\theta_s$, with $\lambda$ following a cosine schedule from 0.996 to 1. The inverse temperature for student classification head, $\tau_s$, is set to $1/0.1$, while the inverse temperature for teacher classification head, $\tau_t$, follows a linear warm-up from $1/0.04$ to $1/0.07$ during the first 30 epochs. For data augmentation, we employ the method from DINO [10] (*e.g.*, color jittering of brightness, contrast, saturation and hue, Gaussian blur and solarization) with preceding random crops and resizing (to 224×224) and make them asymmetric.

The exact settings of augmentation are provided in the next section. Regarding the training cost, when using the ViT-S/16 model and identical hardware configurations, the per-epoch training time of FLSL is 1.19x longer than DINO. Meanwhile, Self-patch's training duration is comparable to FLSL, being 1.21x longer than DINO.

## E.3 Data Augmentation

The augmentation settings in FLSL are based on the augmentation pipeline of DINO [10] with one key modification: the random cropping operation is made asymmetric for the teacher and student networks. In our approach, we begin by sampling two random crops from the input image using a large ratio (*e.g.*, $0.8 \sim 1.0$) at the same location but with different pixel treatments. From each of the crops, we further sample a smaller crop using a ratio of (*e.g.*, $0.5 \sim 1.0$). The smaller crops are then assigned to the student network, while the larger crop are passed to the teacher network. This asymmetry ensures that the queries from the student exist within the teacher's view. Conversely, using symmetric random cropping for both networks adversely affects training performance and leads to collapse. Details of the data augmentation pipeline are listed below. The operations are performed sequentially to produce each view.

- For *Teacher network*, random cropping an area uniformly sampled with a size ratio between $0.8$ to $1.0$, followed by resizing to $224^2$. `transforms.RandomResizedCrop(224, scale=(0.8, 0.1))` in PyTorch.
- For *Student network*, random cropping the crops from teacher network with an area uniformly sampled with a size ratio between $0.5$ to $1.0$, followed by resizing to $224^2$. This results in an effective scale ratio of $(0.4, 1.0)$. `transforms.RandomResizedCrop(224, scale=(0.5, 1.0))` in PyTorch.
- Color jittering of brightness, contrast, saturation and hue, with a probability of $0.8$. `ColorJitter(0.4, 0.4, 0.2, 0.1)` in PyTorch.
- Grayscale with a probability of $0.2$. `transforms.RandomGrayscale(p=0.2)` in PyTorch.
- Gaussian blur with a probability of $0.5$ and uniform random radius from $0.1$ to $2.0$.
- Solarization with a probability of $0.2$.
- Color normalization with mean $(0.485, 0.456, 0.406)$ and standard deviation $(0.229, 0.224, 0.225)$.

## E.4 PyTorch Pseudocode of FLSL

---

**Algorithm 1** FLSL PyTorch Pseudo-code

---

```
# fs, ft:  student and teacher transformer branches
# sa, ca:  self-attention and cross-attention head
# fc:  fully-connected layer
# tp_s, tp_t:  student and teacher inverse temperatures
# a, g, r:  coefficient for the three loss terms
# l:  network momentum rates
ft.params = fs.params
for x in Loader:# load a minibatch x with B samples
      # random augmentation
      x1, x2 = transforms_t(x)
      x1_s, x2_s = transforms_s(x1, x2)
      s1, s2 = fs(x1_s), fs(x2_s)# [B, N, D]
      t1, t2 = ft(x1), ft(x2)# [B, N, D]

      loss = 0.5 * M(s1, t2) + 0.5 * M(s2, t1)
      loss.backward()# back-propagation

      # student and teacher updates
      updates(fs)# SGD
      ft.params = l*ft.params + (1 - l)*fs.params

def H(s, t):
      # s, t:[B, N, D]
      zs, zt = fc(s), fc(t) # [B, N, K]
      ps, pt = softmax(zs/tp_s, dim=-1), softmax(zt/tp_t, dim=-1)
      ps_b = ps.sum(dim=-2).mean(dim=-1)
      return - (pt * log(ps)).sum(dim=-1).mean(),  ps_b*log(ps_b)

def M(s, t):
      t.detach()# stop gradient
      s0 = s.normalize(dim=-1)
      s = sa(s)
      t = ca(s, t, t)
      s0_a = s.normalize(dim=-1)
      h1, h2 = H(s, t)
      ds = ((s0 - s0_a) * (s0 - s_a)).sum(dim=-1).mean()
      return a * ds + g * h1 + r * h2
```

---

Note that a constant $\log K$ (as a result of a uniform prior $\pi = 1/K$) is omitted in the algorithm table. Specifically, with a uniform prior $\pi = 1/K$, the KL divergence term in the objective function (13) reduces to the entropy of the student prediction plus a constant $\log K$, the latter of which is omitted in the algorithm table.

# F  Protocol for hyperparameter tuning

As discussed in the main paper, we need a protocol to evaluate the quality of the learned dense features during the FLSL training for hyperparameter tuning. However, standard evaluation protocols, such as $k$-NN classifier or linear probing are not suitable. We therefore propose a bounding box-aligned $k$-NN classification by leveraging the bounding box information provided by ILSVRC [51].

As shown in Figure 4(a), we partition the bounding box into $s \times s$ grids and find the coordinates of the center for each grid (the green dots). We then locate the $s^2$ features in the feature map, $\hat{\boldsymbol{Z}}$, from the nearest neighbor as shown in Figure 4(b), and store them into the memory bank with label information. For images with multiple bounding box annotations, we pick the largest one. An image is considered correctly classified as long as there is one of the $s^2$ features matching its true category with the prediction. We set $s = 3$ for our training and inflate the number of the nearest neighbors $k$ by a scale factor $c_s$ as the memory bank increases 9 times. We set $k = 20$ and $c_s = 7$ for the best performance.

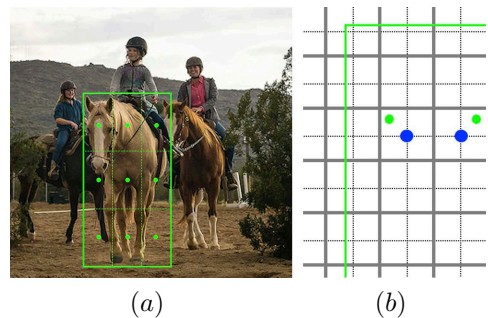

$(a)$ $(b)$

Figure 4: The alignment between bounding box grid centers and the feature centers. We first construct a $3 \times 3$ grid from the bounding box and locate the grid centers. As shown in (a), the 9 grid center points are marked in green. Given the patch size (*e.g.*, $16 \times 16$) for each grid center, we then locate the patch with its center closest to the grid center, as shown in (b).

| Method | Arch. | #params | #epochs | im/s | $k$-NN |
|---|---|---|---|---|---|
| Supervised | RN-50 | 23M | 300 | 1237 | 79.3 |
| *SOTA SSL methods with Big CNNs* | | | | | |
| SwAV | RN50w5 | 586 | 800 | 76 | 67.1 |
| BYOL | RN200w2 | 250 | 1000 | 123 | 73.9 |
| SimCLR-v2 | RN152w3+SK | 794 | 1000 | 76 | 73.1 |
| Supervised | ViT-S/16 | 21M | 300 | 1007 | 79.8 |
| BYOL | ViT-S/16 | 21M | 600 | 1007 | 66.6 |
| MoCov2 | ViT-S/16 | 21M | 600 | 1007 | 64.4 |
| MoCov3 | ViT-S/16 | 21M | 1200 | 1007 | 66.5 |
| SwAV | ViT-S/16 | 21M | 2400 | 1007 | 66.3 |
| iBOT | ViT-S/16 | 21M | 3200 | 1007 | 75.2 |
| DINO | ViT-S/16 | 21M | 3200 | 1007 | 74.5 |
| FLSL | ViT-S/16 | 21M | 1600 | 1007 | 76.7* |
| *Comparison across transformer variants* | | | | | |
| DINO | ViT-B/16 | 85M | 1200 | 312 | 76.1 |
| MoCov3 | ViT-B/16 | 85M | 1200 | 312 | 69.7 |
| EsViT | Swin-S | 49M | 600 | 467 | 76.8 |
| EsViT | Swin-B | 87M | 600 | 297 | 77.7 |
| iBOT | Swin-T | 28M | 1200 | 726 | 75.3 |
| iBOT | ViT-B/16 | 85M | 1600 | 312 | 77.1 |
| iBOT | ViT-L/16 | 307M | 1000 | 102 | 78.0 |
| DINO | ViT-B/8 | 85M | 1200 | 63 | 77.4 |
| DINO | ViT-S/8 | 21M | 3200 | 180 | 78.3 |
| EsViT | Swin-S/W=14 | 49M | 600 | 383 | 77.3 |
| EsViT | Swin-B/W=14 | 87M | 600 | 254 | 78.3 |
| iBOT | Swin-T/W=14 | 28M | 1200 | 593 | 76.2 |
| FLSL | ViT-B/16 | 85M | 600 | 312 | 77.8* |

Table 9: K-NN CLASSIFICATION ON IMAGENET

We present the evaluation results of the bounding box-aligned $k$-NN of FLSL with the standard instance-level $k$-NN of other methods in Table 9. These results provide insights into the global and local semantic coherence of the learned representations. As the bounding box-aligned $k$-NN results in representations with less noise, we mark our results with ($*$) symbol to indicate a **biased** comparison. Note that FLSL is designed for dense prediction tasks and not for instance-level image classification. This Bbox-aligned $k$-NN classification is employed only for hyperparameter tuning and ablation study of the FLSL pipeline.

## G  Transfer learning settings

**MS-COCO setup** We evaluate the performance of the pretrained models on the MS-COCO object detection and instance segmentation tasks with different two-staged frameworks. For ViT-S/16 and ViT-S/8 with Mask R-CNN [28] and FPN [41], we employ multi-scale training following [6] and

resize the image to ensure the short side falls within the range of $480$ to $800$ pixels, while ensuring the long side does not exceed $1,333$ pixels. For a fair comparison, we primarily adhere to the training setting utilized in [75]. Specifically, we employ the AdamW optimizer with a batch size of $16$. Learning rate is linearly warmed up for the first $1,000$ iterations to reach $5e-5$ and subsequently decayed at step $8$ and $11$. Models are trained under 1x schedule. For ViT-B/16 with Mask R-CNN and a simple FPN, we follow the training methodology outlined in Li et al. (2022) [40]. Specifically, the input images are resized to $1,024 \times 1,024$ and augmented with large-scale color jitter ranging from $0.1$ to $2.0$. The model is fine-tuned for 100 epochs using the AdamW optimizer with a weight decay of $0.1$. To adjust the learning rate, we employ a step-wise decay strategy. During the training, the base learning rate is set to $0.0001$, which is gradually increased from $0.0$ to the base rate for the first 250 iterations as a warm-up phase. Additionally, we apply a layer-wise learning rate decay of $0.7$.

**UAVDT setup** The UAVDT dataset contains $23,258$ images for training and $15,069$ images for test. The resolution of the images is about $1,080 \times 540$ pixels. The dataset is acquired with a UAV platform at a number of locations in urban areas. The categories of the annotated objects are car, bus, and truck. The training configuration is adapted from the original setting in [74]. The input size is rescaled to $1,072 \times 528$. The model is trained under 1x schedule. We adopt SGD optimizer with $0.9$ momentum, $0.0001$ weight decay and a batch size of $16$. The base learning rate sets to $0.0005$ with a linear warm-up for the first 300 iterations. The learning rate decreases at the 8th epoch.

# H   ADE20K semantic segmentation

We also evaluate semantic segmentation performances of pre-trained models on ADE20K, which includes 150 fine-grained semantic categories and 25k training data. In line with SelfPatch, all models are fine-tuned with Semantic FPN under the standard 40k iteration schedule, other major settings include input size 512x512, feature layer=[2,5,8,11], Adam optimizer w/ lr=6e-5, poly-scheduler w/ p=1.0, weight decay=0.01 excluding positional embedding and layer norm. Results are reported in table

| Method | Arch | Backbone | #Iter. | mIoU | aAcc | mAcc |
|--------|------|----------|--------|------|------|------|
| MoCo-v2 | FPN | RN50 | 40k | 35.8 | 77.6 | 45.1 |
| SwAV | FPN | RN50 | 40k | 35.4 | 77.5 | 44.9 |
| ReSim | FPN | RN50 | 40k | 36.6 | 78.4 | 46.4 |
| DenseCL | FPN | RN50 | 40k | 37.2 | 78.5 | 47.1 |
| MoCo-v3 | FPN | ViT-S/16 | 40k | 35.3 | 78.9 | 47.1 |
| MoBY | FPN | ViT-S/16 | 40k | 39.5 | 79.9 | 47.1 |
| DINO | FPN | ViT-S/16 | 40k | 38.3 | 79.0 | 47.1 |
| DINO+SelfPatch | FPN | ViT-S/16 | 40k | 41.2 | 80.7 | 52.1 |
| ADCLR | FPN | ViT-S/16 | 40k | 42.4 | 81.1 | 54.2 |
| FLSL | FPN | ViT-S/16 | 40k | **42.9** | **81.5** | **55.1** |

Table 10: **ADE20K** Performances of the recent self-supervised approaches pre-trained on ImageNet-1K. The metrics mIoU, aAcc, and mAcc refer to the mean intersection of union, all pixel accuracy, and mean class accuracy, respectively. FLSL consistently outperforms all the baselines.

# I   Ablation study

## I.1   Impact of batch size

We study the impact of the batch size on the features extracted by FLSL. Table 11 shows that FLSL can achieve high performance with small batch sizes. Unlike the instance-level SSL methods that tend to focus on foreground contents (*e.g.*, objects), FLSL considers all the semantics in an image, *i.e.*, all the features $z$s find their own cluster representatives $\hat{z}$s through the self-attention (*mean-shift*) update. This enriches feature diversity and improves the variance of a mini-batch and benefits the training with small batch sizes.

| Batch size | 64 | 128 | 256 | 512 | 1024 | 2048 |
|---|---|---|---|---|---|---|
| $k$-NN top-1 | 66.1 | 69.8 | 71.7 | 72.4 | 72.4 | 71.9 |

Table 11: IMPACT OF BATCH SIZE

## I.2 Impact of random pooling

In FLSL, contrasting among dense features can be computationally expensive, *i.e.*, $14^2 = 196$ representations to be considered in the objective. Therefore, we apply a random pooling to the queries from the last ViT layer and study the impact of different window sizes of the random pooling.

| Window size | $2 \times 2$ | $4 \times 4$ |
|---|---|---|
| $k$-NN top-1 | 72.4 | 71.1 |

Table 12: IMPACT OF RANDOM POOLING

## I.3 Impact of the number of centroids $K$

We formulate FLSL as an explicit clustering problem. Therefore, the output dimension of the last fully-connected layer is equal to the number of centroids $K$. As shown in Table 13, FLSL enjoys a smaller output dimension compared to its instance-level counterpart, DINO ($K = 65, 536$) [10]. This is mainly due to the higher variance of features in an image than that of a feature cluster. Take ImageNet for instance, the content of an image may range from a single object and stuff to a melange of them from different categories. This requires a large number of centroids to cover the image distribution. While for a semantic cluster, it tends to contain features of high correlation, *e.g.*, features of similar texture, or multiple adjacent objects from the same category, hence requires less centroids to cover its distribution. From the experiment, we find that a large number of centriods improves the performance, but is detrimental and costly when being too large. We pick $K = 4,096$ for all our experiments as it strikes a good balance between performance and cost-effectiveness.

| $K$ | 1024 | 2048 | 4096 | 8192 | 16384 |
|---|---|---|---|---|---|
| $k$-NN top-1 | 68.1 | 72.1 | 72.4 | 72.5 | 72.1 |

Table 13: IMPACT OF NUMBER OF CENTROIDS $K$

## I.4 Ablation on the FLSL objective function

The FLSL objective contains three components: (1) similarity between $\ell_2$-normalized $\boldsymbol{z}$ (features) and $\hat{\boldsymbol{z}}$ (modes), (2) cross-entropy of the probabilities of an augmented pair $H(p(\hat{\boldsymbol{z}}^+), p(\hat{\boldsymbol{z}}))$, and (3) the non-empty constraint $D_{\mathrm{KL}}(\bar{\boldsymbol{p}}\|\boldsymbol{\pi})$:

$$\min \frac{1}{N'} \sum_{\boldsymbol{Z} \in \mathcal{Z}} \sum_{\boldsymbol{z} \in \boldsymbol{Z}} \upsilon \|\boldsymbol{z} - \hat{\boldsymbol{z}}\|_{\mathcal{F}}^2 + \eta \sum_{\boldsymbol{z} \in \boldsymbol{Z}} \mathrm{H}(\boldsymbol{p}(\hat{\boldsymbol{z}}^+), \boldsymbol{p}(\hat{\boldsymbol{z}})) + \gamma D_{\mathrm{KL}}(\bar{\boldsymbol{p}}\|\boldsymbol{\pi}), \tag{31}$$
$$\text{with } \hat{\boldsymbol{z}} = \mathrm{SA}(\boldsymbol{z}, \boldsymbol{Z}, \boldsymbol{Z}), \ \hat{\boldsymbol{z}}^+ = \mathrm{CA}(\boldsymbol{z}, \boldsymbol{Z}^+, \boldsymbol{Z}^+).$$

It is computationally expensive to optimally determine the values of more than two coefficients by performing grid search, especially when the ratios among them are large. We tackle this problem by first fixing $\eta = 1$ and setting $\gamma = 1$ along with the Sinkhorn normalization [19] to perform a grid search on the value of $\upsilon$ with the empirical base condition $\upsilon \leq 1$ and $\gamma \geq 1$ [75, 1]. With the fixed $\upsilon$, we then perform another grid search on $\gamma$ without the Sinkhorn normalization. We implement Sinkhorn normalization [19] as the softmax operation along the batch dimension. Table 5 summarizes the score of $k$-NN evaluation using different coefficient settings. We also visualize the impact of different ratios of the first and second level clustering $\upsilon/\eta$ of the FLSL objective in Figure 5 by visualizing the aggregated similarity score (ASS) map. As the ratio increases, the ASS map shifts from being clear and bright to becoming cluttered and dark. This change occurs because the self-attention for each query becomes more focused, attending to a smaller neighborhood. A smaller ratio leads to larger clusters, which aggregate more attention scores in the region, resulting in a brighter map, particularly in the background. Conversely, a large ratio leads to small, cluttered clusters with fewer attention scores aggregated, resulting in a darker map. A smaller ratio may smooth out small details, while a larger ratio causes the model to focus excessively on local features. From the results in Table 14, a ratio of 0.03 strikes a good balance in between.

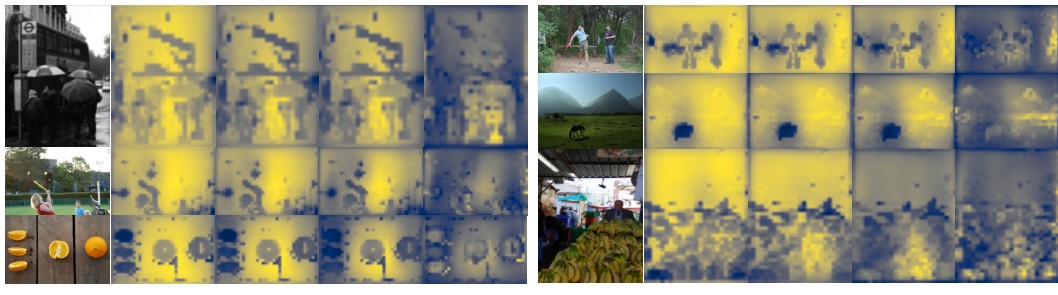

| | image | $v/\eta = .01$ | $v/\eta = .02$ | $v/\eta = .03$ | $v/\eta = .1$ | | image | $v/\eta = .01$ | $v/\eta = .02$ | $v/\eta = .03$ | $v/\eta = .1$ |

Figure 5: Impact of the ratio $v/\eta$ on local semantic consistency with the FLSL-learned representations. The figure presents a visualization of the aggregated similarity scores (ASS) map. As the ratio $v/\eta$ increases, the attention for each query becomes more focused, specifically attending to regions of closer proximity, resulting in more cluttered and smaller dark regions in the ASS map.

| Sinkhorn | $\eta$ | $\gamma$ | $v = 0.0$ | $v = .01$ | $v = .02$ | $v = .03$ | $\sim$ | $v = 0.1$ |
|---|---|---|---|---|---|---|---|---|
| ✓ | 1.0 | 1.0 | 0.1 | 68.7 | 70.7 | 71.2 | $\sim$ | 65.1 |
| ✕ | 1.0 | 1.0 | - | - | - | 66.6 | - | - |
| ✕ | 1.0 | 5.0 | - | - | - | 72.4 | - | - |

Table 14: IMPACT OF THE COEFFICIENTS IN THE FLSL OBJECTIVE.

## J  Aggregated attention score visualizations

To further evaluate the caliber of the learned representations, we contrast the ASS visualization of FLSL with that of DINO, which is a representative instance-level SSL. The input images for this comparison are randomly selected from an ImageNet-1K subset. As shown in Figure 6, the aggregated similarity score (ASS) maps of the tokens from the last layer of a ViT-S/16 trained via FLSL and DINO are visualized and juxtaposed for comparison. For well-clustered tokens at the object-level, the shade distribution of an object in ASS should align with the object shape and be proportional to the object size, i.e., darker shades for smaller objects. To better illustrate, we draw bounding boxes around conspicuous objects in "dimmed" images (in "bboxes" column). Apparently, FLSL leads to ASS better aligned with underlying objects or stuff (e.g., images of "ostrich", "junco", "cowboy boot", "teddy bear", "seashore", "swimming trunks", "sax" etc.), and captures more objects alongside the label-related object in an image (e.g., images of "quail", "plectron", "bloodhound", "tiger shark", etc.), while DINO tends to single out the label-related tokens and drives the tokens in the rest of an image to be highly correlated (e.g., image of "quail", "junco", "blood hound").

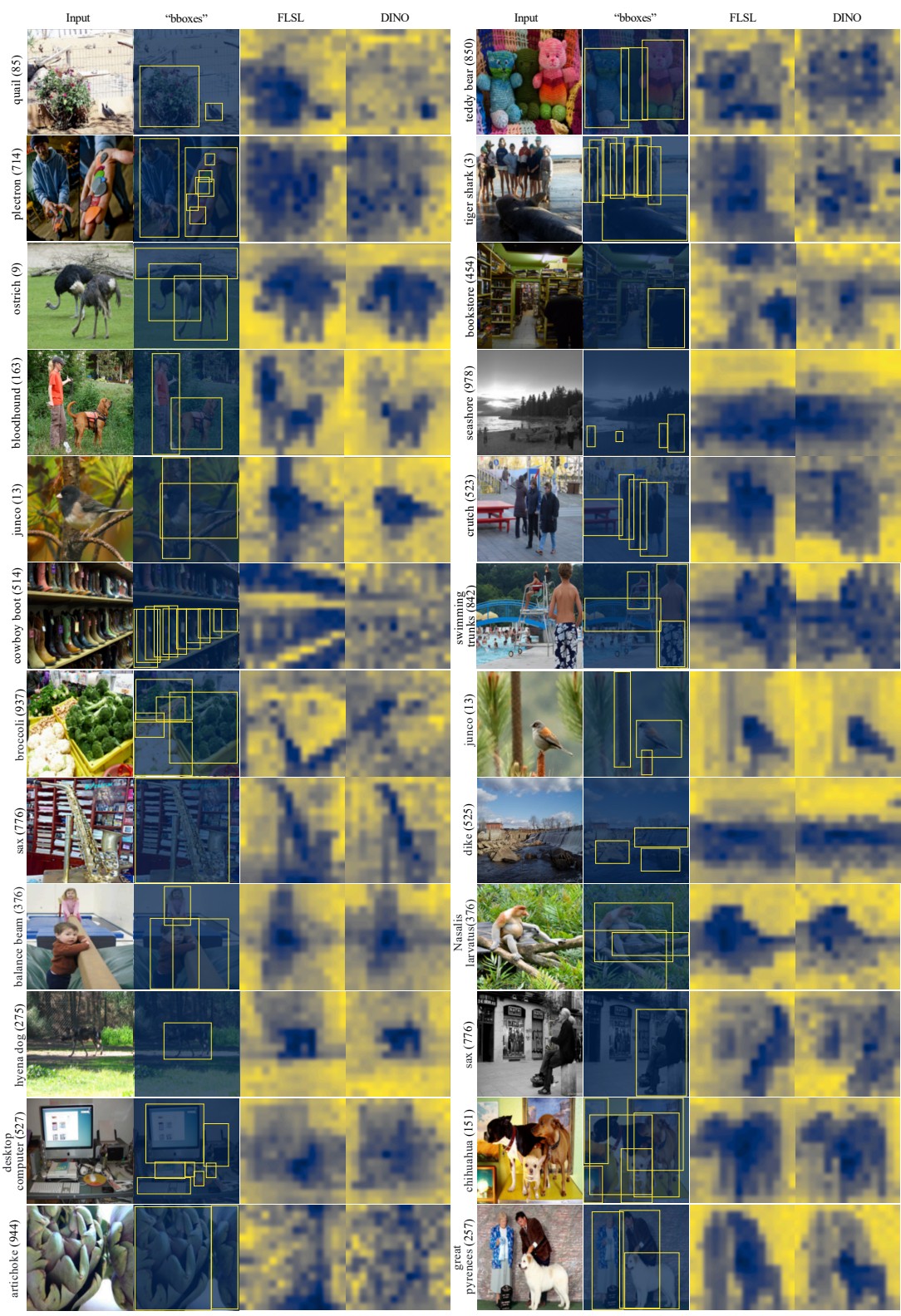

Figure 6: ASS visual comparison between FLSL and DINO