# OpenReview forum: "FLSL: Feature-level Self-supervised Learning"
_NeurIPS.cc/2023/Conference — NeurIPS 2023 poster_

### Official Review · Reviewer_taUM · 2023-06-13

**Soundness:** 3 good
**Presentation:** 3 good
**Contribution:** 3 good
**Rating:** 7
**Confidence:** 5

**Summary:**

This paper proposes Feature-level Self-supervised Learning (FLSL) to handle dense prediction downstream tasks. Specifically, the authors employ the transformer for joint embedding and clustering and construct the objectives from the mean-shift and k-means perspectives. Experiments show that FLSL yields significant improvements in dense prediction tasks.

**Strengths:**

1. The paper is well-motivated and the proposed method seems to work well on dense prediction tasks.

2. The analysis of the connection between mean-shift clustering and SA sounds reasonable. The authors analyze the relationship between ViT and clustering from a new perspective.

3. Experimental results on detection and segmentation show significant improvements and demonstrate the effectiveness of FLSL.

**Weaknesses:**

1. The relation between ADCLR and FLSL is not clear. I see both ADCLR and FLSL use cross-attention to learn patch-level information. So, what's the main difference between ADCLR and the inter-view objective?

2. ''feature-level'' looks somewhat misleading. For me, we typically divide the SSL into ''feature-wise'' (Barlow Twins [1], ZeroCL [2], ARB [3], VICReg [4]) and ''instance-wise'' methods (SimCLR [5], Moco [6]) by the objectives on the different dimension. The proposed method is more like a cluster-level method (SwAV [7]).

[1] Zbontar J, Jing L, Misra I, et al. Barlow twins: Self-supervised learning via redundancy reduction[C]//International Conference on Machine Learning. PMLR, 2021: 12310-12320.

[2] Zhang S, Zhu F, Yan J, et al. Zero-cl: Instance and feature decorrelation for negative-free symmetric contrastive learning[C]//International Conference on Learning Representations. 2021.

[3] Zhang S, Qiu L, Zhu F, et al. Align representations with base: A new approach to self-supervised learning[C]//Proceedings of the IEEE/CVF Conference on Computer Vision and Pattern Recognition. 2022: 16600-16609.

[4] Bardes A, Ponce J, LeCun Y. Vicreg: Variance-invariance-covariance regularization for self-supervised learning[J]. arXiv preprint arXiv:2105.04906, 2021.

[5] Chen T, Kornblith S, Norouzi M, et al. A simple framework for contrastive learning of visual representations[C]//International conference on machine learning. PMLR, 2020: 1597-1607.

[6] He K, Fan H, Wu Y, et al. Momentum contrast for unsupervised visual representation learning[C]//Proceedings of the IEEE/CVF conference on computer vision and pattern recognition. 2020: 9729-9738.

[7]  Caron M, Misra I, Mairal J, et al. Unsupervised learning of visual features by contrasting cluster assignments[J]. Advances in neural information processing systems, 2020, 33: 9912-9924.

**Questions:**

See weakness.

**Limitations:**

No clearly visible limitations.

---

> ### Author Rebuttal · Authors · 2023-08-09
>
> \
> __Weaknesses:__
>
> __1.__  _The relation between ADCLR and FLSL is not clear. I see both ADCLR and FLSL use cross-attention to learn patch-level information. So, what's the main difference between ADCLR and the inter-view objective?_
>
> Thanks for raising this question. The main difference between ADCLR and the FLSL inter-view objective is that ADCLR leverages a specially designed cross-attention (CA) between the pseudo CLS tokens (constructed from local crops) and patch tokens to retain the local information throughout the transformer, while FLSL inter-view objective encourages the consistent representations of positive clusters automatically determined via CA between tokens from student and teacher views. More details of the usage of CA in FLSL and ADCLR are provided below.
>
> * The CA in ADCLR occurs between "pseudo" CLS (pCLS) tokens and patch tokens. The "pseudo" CLS tokens are constructed with the instance-level representations of several small local crops via a dedicated projector, while the CA in FLSL occurs between the output tokens of student ViT and teacher ViT.
>
> *  The CA in ADCLR is uni-directional, i.e., from one pCLS token to patch tokens plus the pCLS itself, while the attention function of patch tokens does not consider pCLS tokens at all. In addition, there is no interaction among different pCLS tokens, whereas the CA in FLSL follows the common CA definition.
>
> *  The CA in ADCLR occurs at every layer in ViT, while the CA in FLSL only occurs at the end of the two ViTs.
>
> We will include this discussion in the appendix.
>
> \
> __2.__ _"feature-level" looks somewhat misleading. For me, we typically divide the SSL into "feature-wise" (Barlow Twins [1], ZeroCL [2], ARB [3], VICReg [4]) and "instance-wise" methods (SimCLR [5], Moco [6]) by the objectives on the different dimension. The proposed method is more like a cluster-level method (SwAV [7])._
>
> Thanks for raising this question. The "level" in our paper refers to the semantic level, at which SSL operates. In this sense, all the above-mentioned methods can be categorized as instance-level since those methods learn meaningful representation for a whole image. FLSL, on the other hand, learns a semantically meaningful representation of a cluster of features, which is determined via mean-shift clustering at the patch/feature-level. Hence, we coined FLSL for our proposed method. We will clarify this further in the introduction section.

---

> > ### Comment · Reviewer_taUM · 2023-08-15
> > **Official Comment by Reviewer taUM**
> >
> > Thank the authors for the detailed response. I tend to accept this submission.

---

### Official Review · Reviewer_Hfeu · 2023-07-03

**Soundness:** 3 good
**Presentation:** 2 fair
**Contribution:** 3 good
**Rating:** 6
**Confidence:** 4

**Summary:**

This paper tackles self-supervised representation learning and primarily forces on learning representations for dense downstream prediction tasks, such as object detection and instance segmentation. To improve upon prior work, the main idea of the paper consists of two parts: (1) the paper leverages the underlying mean shift clustering process via the attention mechanisms of ViTs. In particular, the presented method relies on self-attention and cross-attention layers to achieve intra-level feature clustering. (2) In addition, the presented method takes a k-means perspective to achieve inter-level feature clustering.. The paper demonstrates the effectiveness of the approach via extensive experiments for dense prediction tasks, outperforming prior art for object detection and instance segmentation on MS-COCO.


**Strengths:**

- The paper makes an interesting observation by connecting the attention mechanism (self-attention and cross-attention layers in particular) to mean shift clustering. This results in a relatively clean implementation (see figure 2 for an overview).

- The overall idea is intuitive:
1. The feature representations that belong to a certain cluster are close to the cluster representative (center) and pushed away from the representatives of other clusters.
2. The cluster representatives of the positive areas are pulled together (positives)
This is also reflected in the final loss function and can be implemented via a clean implementation. The pseudo-code in the supplementary materials is very insightful.

- The presented approach demonstrates strong performance on multiple downstream tasks and datasets.  MS-COCO is used for object detection and instance segmentation, UAVDT is used for object detection from UAV platforms, and DAVIS is used for video instance segmentation. The proposed method consistently outperforms DINO (a strong baseline) in these cases.

- The paper contains ablations about the loss function and impact of the number of clusters K. The supplementary also includes additional details about the training setups.

- Overall, the paper is well-written


**Weaknesses:**

- A few comparisons or discussions with related works are missing. For instance, DeepCluster [a, b], PCL [c], and CDL [d] show similarities with the presented approach as they also leverage clustering. A comparison with CDL would be  the most interesting as it relies on instance level and group level losses.

- The presented approach is slightly more complex than prior work (DINO). In particular, the presented approach requires additional attention layers.
As the architecture differs from conventional methods due to the introduction of the self-attention and cross-attention layers, what is the additional computational cost? This information is currently not present in the paper.

- It’s also not clear how robust the method is towards certain dataset biases (e.g., object-centric datasets or imbalanced datasets). The approach relies on a uniform prior over the clusters, which is valid for the ImageNet dataset. However, prior works have been able to pretrain on uncurated datasets (e.g., SEER [e]). It’s currently not clear if this is also possible with the proposed method. It would be valuable to include experiments for COCO pretraining.

[a] Caron et al., Deep clustering for unsupervised learning of visual features, ECCV 2018.

[b] Caron et al., Unsupervised pre-training of image features on non-curated data, ICCV 2019.

[c] Li et al., Prototypical contrastive learning of unsupervised representations, ICLR 2021.

[d] Wang et al., Unsupervised Feature Learning by Cross-Level Instance-Group Discrimination, CVPR 2021.

[e] Goyal et al., Self-supervised Pretraining of Visual Features in the Wild, 2021.


**Questions:**

How well does pretraining work on non-curated datasets, like the COCO or OpenImages datasets?
What is the additional computational cost compared to DINO?


**Limitations:**

The paper briefly mentions the limitations near the end of the paper.

---

> ### Author Rebuttal · Authors · 2023-08-09
>
> \
> __Weaknesses:__
>
> __1.__  _A few comparisons or discussions with related works are missing. For instance, DeepCluster [a, b], PCL [c], and CDL [d] show similarities with the presented approach as they also leverage clustering. A comparison with CDL would be the most interesting as it relies on instance level and group level losses._
>
> Thanks for the suggestion. In the early version of this paper, we had a subsection reviewing the related works in deep clustering, including the above-mentioned ones [a,b,c,d]. However, due to page limit and the scope of the paper, we had to omit them from the submitted version. We believe the connection between FLSL and deep clustering is essential, and we will add the deep clustering subsection back to related work.
>
> \
> __2.__ _The presented approach is slightly more complex than prior work (DINO). In particular, the presented approach requires additional attention layers. As the architecture differs from conventional methods due to the introduction of the self-attention and cross-attention layers, what is the additional computational cost? This information is currently not present in the paper._
>
> The per-epoch training time of FLSL on ViT-S/16 is 1.19x longer than DINO and is on par with SelfPatch, which is 1.21x longer than DINO, under the same model and hardware configuration. We will include this discussion in the main paper.
>
> \
> __3.__ _It’s also not clear how robust the method is towards certain dataset biases (e.g., object-centric datasets or imbalanced datasets). The approach relies on a uniform prior over the clusters, which is valid for the ImageNet dataset. However, prior works have been able to pretrain on uncurated datasets (e.g., SEER [e]). It’s currently not clear if this is also possible with the proposed method. It would be valuable to include experiments for COCO pretraining._
>
> Thanks for the suggestion. We have the following observations regarding the robustness of FLSL towards dataset biases.
>
> * First, ImageNet at the feature/cluster-level can be viewed as uncurated. ImageNet is curated at the instance-level, i.e., balanced for each class. However, ImageNet is single-labelled, and we do not know what the distribution of "class" is when it is on feature/cluster-level or sub-image level, and the number of classes might be way more than 1K. For example, for an image labeled as “hen”, it may contain many objects and stuff alongside a small hen, and some of the objects and stuff in the image may be out of class vocabulary. In this sense, ImageNet can be viewed as an uncurated dataset at the feature/cluster-level (similar to COCO). As shown in the AAS visualization in the general rebuttal pdf file, the inputs are all COCO-like images from ImageNet-1K, FLSL captures non-label-related objects/stuff in the images, which results in more content-aligned AAS, while DINO mostly singles out label-related features and renders the tokens from the rest of the image correlated to each other.
>
> * Second, a uniform prior is a relatively safe choice. For imbalanced dataset, paper [1] shows that a uniform prior is a safer choice as the performance drop as a result of a uniform prior on an imbalanced dataset is much smaller than that of a non-uniform prior (-1.0 vs. -7.2). Thus, for a dataset with agnostic class distribution as in our case, we adopt a uniform prior.
>
> We will include the discussion above in the appendix. Due to restricted time and limited computational resource, we are not able to provide COCO pretrained results for the moment. We will provide these results in the appendix if time permits.
>
> [1] Assran et al., The hidden uniform cluster prior in self-supervised learning, arXiv preprint, 2022.

---

> > ### Comment · Reviewer_Hfeu · 2023-08-12
> > **Questions after rebuttal**
> >
> > I thank the authors for providing the rebuttal. I have 2 additional remarks:
> >
> > 1. As the approach takes ~20% longer during pretraining, I believe it makes sense to include 2 additional baselines: (1) increasing the number of layers in the backbone of DINO by ~20%; (2) increasing the training time of DINO by ~20%.
> >
> > 2. While I appreciate the images in the rebuttal, additional pretraining results on COCO would show that the presented approach can be applied to uncurated datasets (also see prior works mentioned in my original review). In addition, were the presented images randomly selected?

---

> > > ### Author Response · Authors · 2023-08-12
> > >
> > > __1.__ _As the approach takes ~20% longer during pretraining, I believe it makes sense to include 2 additional baselines: (1) increasing the number of layers in the backbone of DINO by ~20%; (2) increasing the training time of DINO by ~20%._
> > >
> > > Thanks for your suggestion. For a fair comparison, existing works often compare the performance of different methods using the same architecture, e.g. ViT-S/16. Therefore, considering time efficiency, instead of modifying the model we still compare with DINO ViT-S/16 trained with 300 epochs, but we will report the performance of FLSL ViT-S/16 trained for 250 epochs (i.e., 250 * 1.2 = 300). The training and evaluation on the downstream tasks may take a couple of days.
> > >
> > > \
> > > __2.__ _While I appreciate the images in the rebuttal, additional pretraining results on COCO would show that the presented approach can be applied to uncurated datasets (also see prior works mentioned in my original review). In addition, were the presented images randomly selected?_
> > >
> > > As the pre-training on COCO can be demanding, e.g., hyperparameter tuning might be necessary due to the difference from ImageNet, and extra time is needed to evaluate on the downstream tasks. We will choose lighter model (ViT-S/16) and smaller number of epochs (200) to show the capability of FLSL on uncurated dataset. Hopefully we will get a decent result by the end of discussion period and report to you.
> > >
> > > As stated in the figure caption, images are randomly sampled from the ImageNet.

---

> > > ### Author Response · Authors · 2023-08-20
> > >
> > > __1.__  _As the approach takes ~20% longer during pretraining, I believe it makes sense to include 2 additional baselines: (1) increasing the number of layers in the backbone of DINO by ~20%; (2) increasing the training time of DINO by ~20%._
> > >
> > > \
> > > Per our previous discussions, we conducted FLSL training using 250 epochs, aligning the training duration with DINO's 300-epoch training schedule. The downstream performance on COCO and ADE20K are reported below.
> > >
> > >
> > >
> > > $~~~~~~~~~~~~~~~~~~~$ Table 1. MASK R-CNN ON COCO Object Detection & Segmentation$~~~~~~~~~~~~~~~~~~~~~~~~~~~~~~~~~~~~~~~~~~~~~~~~~~~$
> > >
> > > \
> > > |   Method - #epoch  |  AP_bbox  |  AP_bbox50  |  AP_bbox75  |  AP_mk $~~~~$ |  AP_mk50  |  AP_m70  |\
> > > |   DINO - 300 $~~~~~~~~~~$ |  40.8 $~~~~~~$ |  63.4  $~~~~~~~~~$ |  44.2   $~~~~~~~~~~~$|  37.3  $~~~~~~~~$ |  59.9 $~~~~~~~$ |  39.5 $~~~~~$ |\
> > > |   FLSL - 250 $~~~~~~~~~~~$ |  44.7 $~~~~~~$ |  65.8  $~~~~~~~~~$ |  48.0   $~~~~~~~~~~~$|  __40.9__  $~~~~~~~~$ |  __64.9__$~~~~~~~~$ |  43.9 $~~~~~$ |\
> > > |   FLSL - 300 $~~~~~~~~~~~$ |  __44.9__ $~~~~~~$ |  __66.1__  $~~~~~~~~~$ |  __48.1__   $~~~~~~~~~~~$|  40.8  $~~~~~~~~$ |  64.7 $~~~~~~~$ |  __44.2__ $~~~~~$ |
> > >
> > >
> > > \
> > > Table 2. Semantic FPN ON ADE20K  Semantic Segmentation
> > >
> > > \
> > > |   Method - #epoch  |$~~~~$ aAcc $~~~$  |$~~~~$ MIoU$~~~~$ |$~~~~$ mAcc$~~~~~$ | \
> > > |   DINO - 300 $~~~~~~~~~$ |  79.0 $~~~~~~~$ |  38.3  $~~~~~~~~~$ |  47.1   $~~~~~~~~~~~$| \
> > > |   FLSL - 250 $~~~~~~~~~~~$ |  81.37 $~~~~~$ |  __42.94__  $~~~~~~~$ |  54.43   $~~~~~~~~~$| \
> > > |   FLSL - 300 $~~~~~~~~~~~$ |  __81.47__ $~~~~~$ |  42.91  $~~~~~~~$ |  __55.06__   $~~~~~~~~~$|
> > >
> > >
> > >
> > > Even with the same training duration, FLSL utilizing 250 epochs still outperforms DINO-300. We will include this result in the appendix.
> > >
> > >
> > > \
> > > __2.__ _While I appreciate the images in the rebuttal, additional pretraining results on COCO would show that the presented approach can be applied to uncurated datasets (also see prior works mentioned in my original review). In addition, were the presented images randomly selected?_
> > >
> > > \
> > > Following the settings in [a], we conducted the experiments to showcase FLSL’s performance on uncurated datasets such as COCO. The results are reported in Table 3 below, alongside the existing COCO-pretrained methods utilizing Mask R-CNN RN50 FPN. FLSL delivers improved performance even with a shorter training schedule (400 epochs), and achieves even better performance with 800 epochs. Please note that due to time constraints we used the default hyperparameters as pretraining on ImageNet-1K. It turns out the default hyperparameters are rather robust and yield decent results of pretraining on COCO. We will include this result in the appendix to demonstrate FLSL’s performance on the uncurated dataset.
> > >
> > > \
> > > $~~~~~~~~~~~~~~~~~~~~~~~~~~~~~$ Table 3. Object Detection and Instance Segmentation Fine-tuned on COCO.$~~~~~~~~~~~~~~~~~~~~~~~~~~~~~$
> > >
> > > \
> > > |   Method $~~~$|Backbone | Data   |  #epoch  |  AP_bbox  |  AP_bbox50  |  AP_bbox75  |  AP_mk $~~~~$ |  AP_mk50  |  AP_m70  |\
> > > |   SimCLR $~~~$ |Backbone|COCO|$~~~~$800$~~~~$|  37.0 $~~~~~~$ |  56.8  $~~~~~~~~~$ |  40.3   $~~~~~~~~~~~$|  33.7  $~~~~~~~~$ |  53.8 $~~~~~~~$ |  36.1 $~~~~~$ |\
> > > |   DenseCL  $~~$|ResNet50|COCO|$~~~~$800$~~~~$|  39.6 $~~~~~~$ |  59.3  $~~~~~~~~~$ |  43.3   $~~~~~~~~~~~$|  35.7  $~~~~~~~~$ |  56.5 $~~~~~~~$ |  38.4 $~~~~~$ |\
> > > |   BYOL $~~~~~~$ |ResNet50|COCO|$~~~~$800$~~~~$|  39.5 $~~~~~~$ |  59.3  $~~~~~~~~~$ |  43.2   $~~~~~~~~~~~$|  35.6  $~~~~~~~~$ |  56.5 $~~~~~~~$ |  38.2 $~~~~~$ |\
> > > |   ORL $~~~~~~~~$ |ResNet50|COCO|$~~~~$800$~~~~$| 40.3 $~~~~~~$ |  60.2  $~~~~~~~~~$ |  44.4   $~~~~~~~~~~~$| 36.3 $~~~~~~~~$ |  57.3 $~~~~~~~$ |  38.9 $~~~~~$ |\
> > > |   FLSL $~~~~~~~$ |ViT-S/16$~~~$|COCO|$~~~~$400$~~~~$|   40.9 $~~~~~~$ |  64.7  $~~~~~~~~~$ |  43.9   $~~~~~~~~~~~$|  38.0  $~~~~~~~~$ |  61.4 $~~~~~~~$ |  39.9 $~~~~~$ |\
> > > |   FLSL $~~~~~~~$ |ViT-S/16$~~~$|COCO|$~~~~$800$~~~~$|   41.7 $~~~~~~$ |  64.7  $~~~~~~~~~$ |  45.5   $~~~~~~~~~~~$|  38.4  $~~~~~~~~$ |  62.0 $~~~~~~~$ |  41.0 $~~~~~$ |
> > >
> > > \
> > > \
> > > [a] Xie, J., Zhan, X., Liu, Z., Ong, Y.S. and Loy, C.C., 2021. Unsupervised object-level representation learning from scene images. Advances in Neural Information Processing Systems, 34, pp.28864-28876.

---

### Official Review · Reviewer_asnD · 2023-07-05

**Soundness:** 3 good
**Presentation:** 2 fair
**Contribution:** 2 fair
**Rating:** 5
**Confidence:** 4

**Summary:**

Current self-supervised learning (SSL) methods, including SimCLR, DINO, VICReg, and MOCOv3, focus mainly on instance-level representations, limiting their use in tasks like object detection and segmentation. To overcome this, a new two-level feature clustering SSL method named Feature-Level Self-supervised Learning (FLSL) has been introduced, which uses the mean-shift clustering process of Vision Transformers to improve semantic cluster representations. Experimentally, FLSL outperforms existing SSL methods in dense prediction tasks, delivering impressive results on multiple benchmarks, notably MS-COCO, UAVDT, and DAVIS 2017 datasets.

**Strengths:**

1.It introduces the mean-shift and k-means for the pre-training of dense predictions.
2. The performance is improved with the proposed method.

**Weaknesses:**

1. The idea of considering similar pixels or patches as positive pairs is not new. Many previous works have explored this idea already, e.g. [1].
2. Many related works are not compared, e.g. [2, 3]. There are also some other works that I do not mention.







[1] Hénaff O J, Koppula S, Alayrac J B, et al. Efficient visual pretraining with contrastive detection[C]//Proceedings of the IEEE/CVF International Conference on Computer Vision. 2021: 10086-10096.
[2] Xie Z, Lin Y, Zhang Z, et al. Propagate yourself: Exploring pixel-level consistency for unsupervised visual representation learning[C]//Proceedings of the IEEE/CVF Conference on Computer Vision and Pattern Recognition. 2021: 16684-16693.
[3]  Xiao T, Reed C J, Wang X, et al. Region similarity representation learning[C]//Proceedings of the IEEE/CVF International Conference on Computer Vision. 2021: 10539-10548.

**Questions:**

The authors claimed that they demonstrate for the first time the connection between the attention mechanism mean-shift clustering. However, there are already some works that demonstrate the connection between attention and clustering, e.g. [1]. Does this paper bring any new insights?

[1] Zhou D, Yu Z, Xie E, et al. Understanding the robustness in vision transformers[C]//International Conference on Machine Learning. PMLR, 2022: 27378-27394.

**Limitations:**

The authors have addressed the limitations.

---

> ### Author Rebuttal · Authors · 2023-08-09
>
> \
> __Weaknesses:__
>
> __1.__ _The idea of considering similar pixels or patches as positive pairs is not new. Many previous works have explored this idea already, e.g. [1]._
>
> Thanks for bringing [1] to our attention. Yes, FLSL belongs to the family of SSL methods that consider similar pixels or patches as positive pairs. We have discussed in the related work section the representative SSL methods in this area, i.e., SoCo, ORL, PixPro, LC-loss, SelfPatch+DINO, and ADCLR, etc.
>
> The work in [1] relies on dedicated algorithms (e.g., FH or MCG) to determine the mask of similar pixels/patches, which is analogous to SoCo and ORL that leverage non-trainable _selective search_ algorithms to find the RoIs as positive pairs. In contrast, FLSL is end-to-end trainable. It does not relies on any dedicated non-trainable algorithm for cluster determination. In contrast, FLSL leverages mean-shift clustering as self- and cross-attention to automatically find a positive pair of soft clusters. This accounts for one of our main contributions.
>
> We will incorporate [1] to the related work section, and clarify our contributions further.
>
> \
> __2.__ _Many related works are not compared, e.g. [2, 3]. There are also some other works that I do not mention._
>
> Thanks for pointing out these related works. Indeed, there are many related works on this topic, and we incorporated the most representative ones in the paper, including SoCo, ORL, PixPro, LC-loss, SelfPatch+DINO, and ADCLR, etc. Paper [2], PixPro, has been discussed in related work. Paper [3] takes a contrastive learning strategy to maximize the similarity of global representations and the sliding-window-pooled representations in the overlapped region of two augmented views. There are several existing works considering this joint consistency on views of image and local patches (e.g., DetCo, which has been discussed in our paper). We will include paper [3] in related work too.
>
> \
> __Questions:__
>
> __1.__  _The authors claimed that they demonstrate for the 1st time the connection between the attention mechanism and mean-shift clustering. However, there are already some works that demonstrate the connection between attention and clustering, e.g. [1]. Does this paper bring any new insights?_
>
> Thanks for pointing out this related work. Paper [1] tries to interpret the attention mechanism from the perspective of information bottleneck (IB), which is further related to clustering. From this perspective, paper [1] is relevant to ours.
>
> However, the formulation of IB in paper [1] is essentially an EM-fitting of a GMM with soft assignment under certain major assumptions (i.e., Gaussian approximation in KL minimization and small smoothing scale as shown in their proof provided in the appendix). In contrast with soft GMM, mean-shift clustering is non-parametric (KDE) and poses no priori assumption of the underlying clusters (e.g., unknown number of clusters, no KL minimization involved). This makes mean-shift clustering closer in line with the attention mechanism that does not make much assumption on the input. We will cite paper [1] and include this discussion in related work.

---

> > ### Comment · Reviewer_asnD · 2023-08-15
> >
> > The authors have addressed my concerns.

---

### Official Review · Reviewer_rt9u · 2023-07-07

**Soundness:** 3 good
**Presentation:** 3 good
**Contribution:** 2 fair
**Rating:** 5
**Confidence:** 2

**Summary:**

The authors point out the limitations of previous SSL methods on dense prediction tasks, because of tis instance-level objectives. On the other hand, recent studies focusing on  dense prediction based on region, patch, and pixel to learn globally semantic representations on these sub-regions. To this end, the paper introduces a novel method that learns representations with both local and global semantics by leveraging the mean-shift clustering. The proposed method consists of intra-view clustering with mean-shift and inter-view clustering with k-means. The extensive experimental results shows its effectiveness on various dense prediction tasks.


**Strengths:**

* Well-written paper with superior performances
* Conducted experiments on various dense prediction tasks


**Weaknesses:**

* Some SSL methods that are strong in dense prediction tasks are omitted from the tables. (e.g., iBOT and RC-MAE)
    * For example, iBOT with ViT-S/16 outperform the methods in Table 1 in terms of AP^bbox and AP^mask. RC-MAE is also comparable to iBOT.
* The algorithm table in the appendix seems to have a gap with the equations of the proposed method. Is there some omitted explanation in the algorithm table?
* It would be better to compare the AAS of the method with other state-of-the-art methods so that the qualitative results can visualize distinctive behaviors.

* Minor
    * It would be better for the highest values in the tables to be bold.

[1] Jinghao Zhou, Chen Wei, Huiyu Wang, Wei Shen, Cihang Xie, Alan Yuille, and Tao Kong. "ibot: Image bert pre-training with online tokenizer," ICLR'22

[2] Youngwan Lee, Jeffrey Willette, Jonghee Kim, Juho Lee, Sung Ju Hwang, "Exploring The Role of Mean Teachers in Self-supervised Masked Auto-Encoders," ICLR'23.

**Questions:**

* Is the proposed method also competitive in terms of throughput?
* The scales of random resized crop for each networks seem to have high lower bound compared to previous SSL methods. Is there any reason that the lower bound of scales should be relatively high?
* Does the FLSL also show superiority in the fine-tuning task on ImageNet-1K?

**Limitations:**

The potential limitations are addressed well.

---

> ### Author Rebuttal · Authors · 2023-08-09
>
> \
> __Weaknesses:__
>
> __1.__ _Some SSL methods that are strong in dense prediction tasks are omitted from the tables. (e.g., iBOT and RC-MAE). For example, iBOT with ViT-S/16 outperform the methods in Table 1 in terms of $AP^{bbox}$ and $AP^{mask}$. RC-MAE is also comparable to iBOT._
>
> Thanks for pointing out these related works. We discussed iBOT in the related work section. We choose ViT-S/* as backbone of FLSL and Mask R-CNN as detector because (1) this benchmark has a lower computational cost, and (2) there are more baselines to compare with. We did not include iBOT results in Table 1 because iBOT employs the **Cascade** Mask R-CNN as detector, which is more complex and expensive than Mask R-CNN, leading to an unfair comparison. RC-MAE only provides the results of ViT-B/16 with Mask R-CNN, and we will include its results in Table 2.
>
> \
> __2.__ _The algorithm table in the appendix seems to have a gap with the equations of the proposed method. Is there some omitted explanation in the algorithm table?_
>
> A constant $\log K$ (as a result of a uniform prior $\pi=1/K$) is omitted in the algorithm table. Specifically, with a uniform prior $\pi=1/K$, the KL divergence term in the objective function (13) reduces to the entropy of the student prediction plus a constant $\log K$, the latter of which is omitted in the algorithm table.
>
> \
> __3.__ _It would be better to compare the AAS of the method with other state-of-the-art methods so that the qualitative results can visualize distinctive behaviors._
>
> This is an excellent suggestion. We provided a side-by-side comparison of AAS visualization between FLSL and DINO in the general rebuttal pdf file. As we can see, FLSL leads to AAS better aligned with underlying objects and stuff, and captures more objects alongside the label-related object in an image, while DINO tends to single out the label-related tokens and drives the tokens in the rest of an image to be highly correlated. We will add this visualization and discussion to the appendix.
>
> \
> __4.__ _It would be better for the highest values in the tables to be bold._
>
> We highlighted the FLSL results in light blue in the tables. As suggested, we will highlight the highest values with boldface for better visibility.
>
> \
> __Questions:__
>
> __1.__ _Is the proposed method also competitive in terms of throughput?_
>
> The per-epoch training time of FLSL on ViT-S/16 is 1.19x longer than DINO and is on par with SelfPatch, which is 1.21x longer than DINO, under the same model and hardware configuration. We will include this discussion in the main paper.
>
> \
> __2.__ _The scales of random resized crop for each network seem to have high lower bound compared to previous SSL methods. Is there any reason that the lower bound of scales should be relatively high?_
>
> We set a higher lower-bound to include more contextual information in each view to help the model to learn representations at higher semantic level. Specifically, in FLSL, a positive cluster representation is retrieved via mean-shift cluster assignment in the form of cross-attention. Consider a query token from a source local crop that is very small and only contains a shade of a single color and no structured features, the cross-view cluster assignment of that query would result in a cluster of all patches of the similar color in the target view. This restricts the semantic level to colors and hinders the model from learning meaningful representations at higher semantic level that better aligns with objects or stuff. Therefore, a higher lower-bound can provide more contextual information to facilitate FLSL to learn meaningful representations. We will clarify this further in Appendix 4.
>
> \
> __3.__ _Does the FLSL also show superiority in the fine-tuning task on ImageNet-1K?_
>
> We did not consider fine-tuning on classification task because FLSL is designed mainly for dense prediction tasks and there is no [class] token involved. As a future work, we will explore ways to extend FLSL for tasks that necessities a global representation while retaining its existing properties.

---

### Official Review · Reviewer_ggdQ · 2023-07-26

**Soundness:** 3 good
**Presentation:** 3 good
**Contribution:** 3 good
**Rating:** 7
**Confidence:** 3

**Summary:**

This paper presents FLSL a self-supervised learning model designed to give good performances of the pre-trained model on downstream dense prediction tasks. The proposed method can be summarised as a two-level clustering problem to achieve this objective :

- Intra-view clustering: which aims to cluster together semantically related embeddings and pull apart those which are not. This is achieved using cluster representative which are obtained using mean-shift clustering. The embeddings are then drawn to their representatives. The paper demonstrated the relation between mean-shift and self-attention which is used of that purpose

- Inter-view clustering: the goal is facilitate the representations of semantically similar features across the dataset to be close. This is done by applying a soft-assignment of the representatives obtained previously to a set centroids.



**Strengths:**

1. The demonstration of the relation between mean-shift and self-attention is a strong contribution of this paper and it plays a key role in the intra-view clustering objective
2. The proposed method outperforms other SSL frameworks on standard dense benchmarks(ms-coco,  uavdt, davis)
3. The paper is well written and the authors provided proofs for their claims as well as code and implementation details for reproducibility.

**Weaknesses:**

1. This paper uses a bbox-aligned k-NN classifier to perform the ablation studies of their work, but we don't know how the performances on this classifier correlates with the results on downstream tasks segmentation or detection.
2. It would have been good to have further evaluations on semantic segmentation on other datasets such as ADE20k, PascalVOC, cityscapes to get a broader view of the performances of the model

**Questions:**

1. what is the computational cost of the proposed method compared to other SSL frameworks?
2. Did you evaluate the importance of the different levels of clustering? like how much each participate to the actual benefits on downstream performances. I see some results in Table 5., but it shows the performance with k-NN and always considers the inter-view clustering. What happens when we do not consider the inter-view clustering?
3. Also noticed that using smaller patches on the backbone in FLSL (e.g. Vit-s/8) seems to be better compared to backbone with larger patches(16), do you know the reasons behind this behaviour? (in table 1 and 3)

**Limitations:**

no particular limitations which need to be addressed. The main limitations of this work can be the complexity of the method which the authors spoke about and the potentially computational cost of the framework.

---

> ### Author Rebuttal · Authors · 2023-08-09
>
> \
> __Weaknesses:__
>
> __1.__ _How the performance on bbox-aligned k-NN classifier correlates with dense prediction tasks?_
>
> Thanks for raising this question. FLSL learns dense semantic representations rather than a single instance-level representation. Therefore, we design a bbox-aligned k-NN classifier to evaluate the feature quality for hyperparameter tuning and ablation study.  As detailed in Appendix 5, to make this classifier correlates with the results on downstream tasks of detection or segmentation, the bounding box information provided by ImageNet-1K is leveraged during validation. Specifically, to have a higher bbox-aligned k-NN accuracy, the FLSL-learned representations in the region of ground-truth bounding-boxes should be both locally and globally semantic, such that
> the extracted local features are indeed from the same object in an image while being close to the bbox-aligned features (of the same category) extracted by FLSL in training images for k-NN classification. Our empirical study also shows the effectiveness of this bbox-aligned k-NN classifier for downstream tasks of detection and segmentation. We will clarify this further in Appendix 5.
>
> \
> __2.__ _It would have been good to have further evaluations on semantic segmentation on other datasets such as ADE20k, PascalVOC, cityscapes to get a broader view of the performances of the model._
>
> Thanks for the suggestion. Please see the general rebuttal pdf file, where we provided ADE20k semantic segmentation results of FLSL compared with baseline methods. In line with SelfPatch, all models are fine-tuned with Semantic FPN under the standard 40k iteration schedule. Similar to the results on COCO object segmentation and DAVIS instance segmentation, FLSL consistently outperforms all the baseline methods on ADE20k. We will include this result in the appendix.
>
> \
> __Questions:__
>
> __1.__ _What is the computational cost of the proposed method compared to other SSL frameworks?_
>
> The per-epoch training time of FLSL on ViT-S/16 is 1.19x longer than DINO and is on par with SelfPatch, which is 1.21x longer than DINO, under the same model and hardware configuration. We will include this discussion in the main paper.
>
> \
> __2.__ _Did you evaluate the importance of the different levels of clustering? like how much each participate to the actual benefits on downstream performances. I see some results in Table 5., but it shows the performance with k-NN and always considers the inter-view clustering. What happens when we do not consider the inter-view clustering?_
>
> Yes, Table 5 provides the ablation study of the importance of the different levels of clustering. We always consider inter-view clustering (e.g., $\eta=1$) because it governs the global meaningfulness of the representations. Without considering the inter-view clustering, the 2nd half of the FLSL pipeline in Figure 2 is discarded and the intra-view clustering alone leads to collapse in training and cannot learn semantically meaningful representations. We will clarify this in the caption of Table 5.
>
> \
> __3.__ _Also noticed that using smaller patches on the backbone in FLSL (e.g. Vit-s/8) seems to be better compared to backbone with larger patches (16), do you know the reasons behind this behaviour? (in Table 1 and 3)_
>
> This is because smaller patches results in higher resolution of feature maps, which benefits not only dense prediction tasks but also classification (see e.g., [1]).
>
> [1] Caron et al., Emerging properties in self-supervised vision transformers, ICCV 2021.

---

### Author Rebuttal · Authors · 2023-08-09

Please find the attached pdf file that includes (Figure 1) AAS visual comparison between FLSL and DINO, and (Table 1) ADE20K semantic segmentation results.

---

### Decision · Program_Chairs · 2023-09-21

**Decision:**

Accept (poster)

**Comment:**

This paper proposes a novel self-supervised learning algorithm geared towards dense prediction vision tasks. It does that by introducing a mechanism to learn representations with both local and global semantics.
The paper received generally positive scores, with all reviewers recommending acceptance and praising the clarity of the presentation, the clear motivation, the innovative aspect of the proposed method, and the extensive experimental validation, which demonstrate significant improvement on detection and segmentation tasks over competing baselines.